# An automated single-molecule FRET platform for high-content, multiwell plate screening of biomolecular conformations and dynamics

Andreas Hartmann [1] ✉, Koushik Sreenivasa [1,5,7], Mathias Schenkel [1,7], Neharika Chamachi [1,7], Philipp Schake [1,7], Georg Krainer [1,2,6] & Michael Schlierf [1,3,4] ✉

Single-molecule FRET (smFRET) has become a versatile tool for probing the structure and functional dynamics of biomolecular systems, and is extensively used to address questions ranging from biomolecular folding to drug discovery. Confocal smFRET measurements are amongst the widely used smFRET assays and are typically performed in a single-well format. Thus, sampling of many experimental parameters is laborious and time consuming. To address this challenge, we extend here the capabilities of confocal smFRET beyond single-well measurements by integrating a multiwell plate functionality to allow for continuous and automated smFRET measurements. We demonstrate the broad applicability of the multiwell plate assay towards DNA hairpin dynamics, protein folding, competitive and cooperative protein–DNA interactions, and drug-discovery, revealing insights that would be very difficult to achieve with conventional single-well format measurements. For the adaptation into existing instrumentations, we provide a detailed guide and open-source acquisition and analysis software.

Single-molecule Förster resonance energy transfer (smFRET) has become a widely used technique to monitor biomolecular conformations and dynamics on the nanometer scale[1,2]. Advanced data analysis approaches to characterize dynamics from nanoseconds to minutes and hours[3–7], together with standardizations across many laboratories and open science initiatives, have pushed smFRET to a routinely accessible technique in biophysical and biochemical research[8–13]. smFRET, due to its versatility and sensitivity, is now extensively used to address a wide range of questions in dynamic structural biology and biophysics, including the functional mechanisms of enzymes and membrane proteins[14–16], protein–nucleic acid and small-molecule–protein interactions[17,18], and protein or nucleic-acid folding[8,19,20], to name but a few[2,10,21].

Despite their popularity, smFRET studies often require the curation of large datasets with high statistics and the sampling of a wide parameter space, for example, by varying one or more constituent molecular components in ten or more steps. For instance, structure determination of biomolecular conformations using trilateration

[1]B CUBE Center for Molecular Bioengineering, TU Dresden, Tatzberg 41, 01307 Dresden, Germany. [2]Centre for Misfolding Diseases, Yusuf Hamied Department of Chemistry, University of Cambridge, Lensfield Road, CB2 1EW Cambridge, UK. [3]Physics of Life, DFG Cluster of Excellence, TU Dresden, 01062 Dresden, Germany. [4]Faculty of Physics, TU Dresden, 01062 Dresden, Germany. [5]Present address: Department of Bionanoscience, Delft University of Technology, 2629HZ Delft, Netherlands. [6]Present address: Institute of Molecular Biosciences, University of Graz, Humboldtstrasse 50/III, 8010 Graz, Austria. [7]These authors contributed equally: Koushik Sreenivasa, Mathias Schenkel, Neharika Chamachi, Philipp Schake. ✉e-mail: andreas.hartmann2@tu-dresden.de; michael.schlierf@tu-dresden.de

approaches requires measurement of multiple directions by FRET, which are often probed in combination with a variation of solution conditions[22,23]. To uncover biomolecular interactions and changes in protein and nucleic-acid structures, for example, in functional investigations or folding studies, ligand or co-solute concentrations are typically varied over several orders of magnitude, giving insights into changes in molecular conformations and kinetics as well as information on binding stoichiometries and affinities[24–27]. Such measurements also often demand a high sampling density to alleviate over-parameterization effects and to increase fitting accuracy and precision. Moreover, for the determination of experimental variability, a need for technical and biological replicates arises in general, not least to ascertain reproducibility and enhance scientific rigor[9,13,28]. Hence, rapidly more than 50 conditions need to be probed in smFRET experiments.

Unfortunately, curation of large datasets and sampling of multiple conditions is laborious and time consuming with current smFRET modalities. Confocal smFRET measurements, for example, which are amongst the widely used smFRET assays, are typically performed in a single-well format and measurements are normally conducted in a manual manner, in that, an experimenter needs to replenish and equilibrate the sample after each experiment. Such manual experimentation is also susceptible to changes in environmental conditions, like changes in temperature, instrumental stability, or other parameters. Approaches, based on microfluidic mixing[29] or multi-spot confocal readouts[30], have been devised to address these issues, for example, by providing the possibility to vary constituent molecular concentrations or reducing the measurement times per sample chamber. However, these approaches rely on extensive customization and custom-made hardware, are often not easily integratable into existing setups, and typically require expert knowledge not necessarily available in standard lab settings. Moreover, automation is not easily achievable with these approaches.

An attractive, alternative way to address the need for generating large datasets and sampling of a large parameter space are multiwell plates. Multiwell plates are ubiquitous tools in all areas of science because they allow collecting data for tens to hundreds of different conditions[31]. Hence, implementing multiwell readouts in smFRET experiments should lend itself a powerful approach to probe many different conditions in a fully automated manner within a single continuous experiment under controlled conditions. Fluorescence microscopy experiments for large-scale screening, for example, of single-molecule fluorescence in situ hybridization (smFISH), RNA interference or organoids, have been automated already in many applications based on 96-well or larger multiwell plates[32–34]. A recent study described high-throughput fluorescence correlation spectroscopy for 96-well plates[35], however, to our knowledge the powerful platform of multiwell plates has not yet been transferred to a format suitable for applications in smFRET experiments, neither by manufacturers of microscopes nor by manufacturers of specialized single-molecule spectroscopy instruments.

Here, we introduce an automated smFRET platform for high-content, multiwell plate screening of biomolecular conformations and dynamics. We describe the implementation of multiwell plates for fully automated confocal smFRET measurements in a single, continuous experiment. We provide an open-source software suite for data acquisition, processing, analysis, and visualization. To illustrate the broad applicability of the multiwell plate measurement format, we validate the approach using an array of systems with increasing complexity. Using a DNA ruler system, we show that high precision and accuracy between sample wells can be achieved down to Ångström distances. We further evaluate the possibility to extract millisecond transition rates for DNA nanostructures. In protein-unfolding experiments, we determine thermodynamic stability parameters and parameters related to protein dynamics with high accuracy and sampling

density and gain new insights into protein folding mechanisms. We then expand the multiwell plate system to study multi-component systems by probing the competition of two proteins for the same DNA substrate. We thereby discover a simultaneous binding interaction of RecA and SSB to single-stranded DNA, through the fine sampling in our multiwell plate smFRET assay. Finally, we illustrate the capability to use the multiwell plate smFRET format to screen for small-molecule–protein interactions using a misfolding model of the human cystic fibrosis transmembrane conductance regulator (CFTR) and gain quantitative readouts of the drug–protein interactions. Taken together, we anticipate that our approach will transform smFRET measurements by enabling the acquisition of high-content smFRET datasets for single-molecule analysis in dynamic structural biology and biophysics, and beyond.

## Results

### Automated multiwell plate smFRET measurements

Typical commercial or custom-built confocal smFRET instruments allow for individual experiments in a single-well chamber mounted onto a stage for positioning and focusing. Such single-well chambers require cleaning, refilling, and equilibration for each experiment, making it laborious for the experimenter to conduct measurements, thereby limiting throughput, and affecting potentially stable conditions for a large set of experiments. Here, we extend the capabilities of confocal smFRET beyond single-well measurements by integrating a multiwell plate functionality into a confocal microscope to allow for continuous and automated smFRET measurements (Fig. 1).

To this end, we equipped a multiparameter single-molecule detection microscope with three major additional and commercially available components (Fig. 1, Supplementary Table 1): (i) a motorized x-y stage, capable of holding a multiwell (e.g., 96-well) plate with a position accuracy <40 μm; (ii) a heating pad for the multiwell plate to prevent condensation on the well-plate sealing, which is operated a few Kelvin above the desired temperature (e.g., room temperature); and (iii) a liquid dispenser, which frequently replaces the evaporating immersion medium of the high numerical aperture water objective. During continuous measurements, the confocal detection volume is maintained 30 μm in solution using an autofocus system integrated in the microscope (Supplementary Table 1).

A custom-written, open-source data acquisition software (Fig. 1), written in Python, synchronizes x-y positioning with data acquisition using predefined libraries of the hardware. Using our software's graphical user interface (GUI; pyMULTI; available on GitHub), measurement wells and times including descriptive text can be defined. During measurements, the acquired fluorescence data is saved in subfolders for subsequent data analysis. Data analysis is performed by our open-source GUI software (pyBAT and pyVIZ; available on GitHub) and involves fluorescence burst search routines, smFRET analysis, and data visualization, all following state-of-the-art analysis protocols[9]. In summary, the presented instrument platform with smFRET data acquisition and analysis scripts allows us to perform multiwell plate smFRET experiments in an automated fashion.

In a first set of experiments, we aimed at evaluating the accuracy and precision of multiwell plate measurements and assessing their variability over time. To this end, we performed 96 independent but identical smFRET measurements of a mixture of two rigid double-stranded DNA ruler constructs with 9- and 21-base pair (bp) distance between the acceptor and donor fluorophores, respectively (Fig. 2b, Supplementary Table 4). We chose the two DNA constructs to validate the accuracy and precision in extracting low- and high-FRET efficiency populations, according to published standards[9]. Furthermore, we used the two FRET populations to determine correction factors for each measurement repeat, and thereby assessed the measurement stability (Supplementary Methods and Supplementary Fig. 4). We loaded the mixture into each well of a 96-well plate, and performed smFRET

measurements, using pulsed-interleaved excitation (Fig. 1), a widely used excitation scheme in single-molecule fluorescence spectroscopy[36,37] (see "Methods"). This allowed us to extract additional information for each detected molecule, such as about stoichiometry and lifetime of the donor and acceptor fluorescence. In total, we recorded 20-min-long photon streams of donor and acceptor fluorescence for each of the 96 wells, resulting in a total measurement duration of about 32 h. The data were analyzed using pyBAT and pyVIZ yielding FRET efficiency ($E$) histograms with an average number of $\langle N \rangle = 1322 \pm 108$ bursts per well without any significant loss of molecules over the 32-h-measurement period (Supplementary Fig. 4). We evaluated the measured $E$ values of the two DNA rulers for their accuracy and precision. To this end, we fitted all individual 96 $E$-histograms with two Gaussian distributions to extract the mean FRET efficiencies, the standard deviation, and the ratio of molecules in the 21-bp population (Supplementary Fig. 4C). Remarkably, we found only minimal deviations between the 96 measurements. The cumulated $E$-histogram of all 96 wells (Fig. 2c, top panel) yielded $\left\langle E_{9\text{bp}} \right\rangle = 0.797 \pm 0.001$ and $\left\langle E_{21\text{bp}} \right\rangle = 0.146 \pm 0.001$, which agrees very well with the expected FRET efficiencies ($E_{9\text{bp}}^{\text{theo}} = 0.777 \pm 0.009$ and $E_{21\text{bp}}^{\text{theo}} = 0.159 \pm 0.008$) using accessible volume simulations[38]. The mean absolute deviation is <8.2% (i.e., 0.8 Å for the 9 bp ruler and 1.1 Å for the 21 bp ruler). We found a very low variation of the measured FRET efficiencies between the individual measurements, as gauged by a standard deviation of $\sigma_{\langle E \rangle} < 0.006$. Notably, the minimal loss of molecules throughout the measurement (Supplementary Fig. 4A) as

well as the low variability in the population ratio (Supplementary Fig. 4D) confirms the measurement stability of the assay. Further, we did not observe any significant loss due to photo bleaching. In conclusion, our multiwell plate smFRET platform enables high-fidelity measurement of FRET samples with little variability over time. Uniquely, the multiwell platform provides the possibility to acquire data from tens to hundreds of conditions with high accuracy and precision, in accordance with conventional single-well chamber smFRET measurements[9].

## Probing conformational changes of biomolecules by multiwell plate smFRET

Dynamic structural changes of nucleic acids and proteins are central to their functionalities[2]. smFRET is frequently used to gain structural and dynamic insights into conformational changes of nucleic acids and proteins, and to decipher their folding and assembly mechanisms[8,10,13]. In particular changes of solution conditions are an important tool to tune kinetics of structural transitions and to drive molecules into desired conformations. Here, a large dataset is particularly advantageous to extract kinetic and thermodynamic parameters with high accuracy and precision. To demonstrate this, we applied our multiwell plate smFRET assay to a highly dynamic DNA hairpin system and to the unfolding of a small globular protein.

In a first experiment, we determined the effects of salt on the kinetics of a dynamic DNA hairpin ($hpT_5$) comprising a 5-bp long complementary annealing stem and a single-stranded loop of 21 thymidine (dT) nucleotides (nts) (Fig. 2d inset, Supplementary Table 4). The hairpin was designed such that it transiently anneals at a

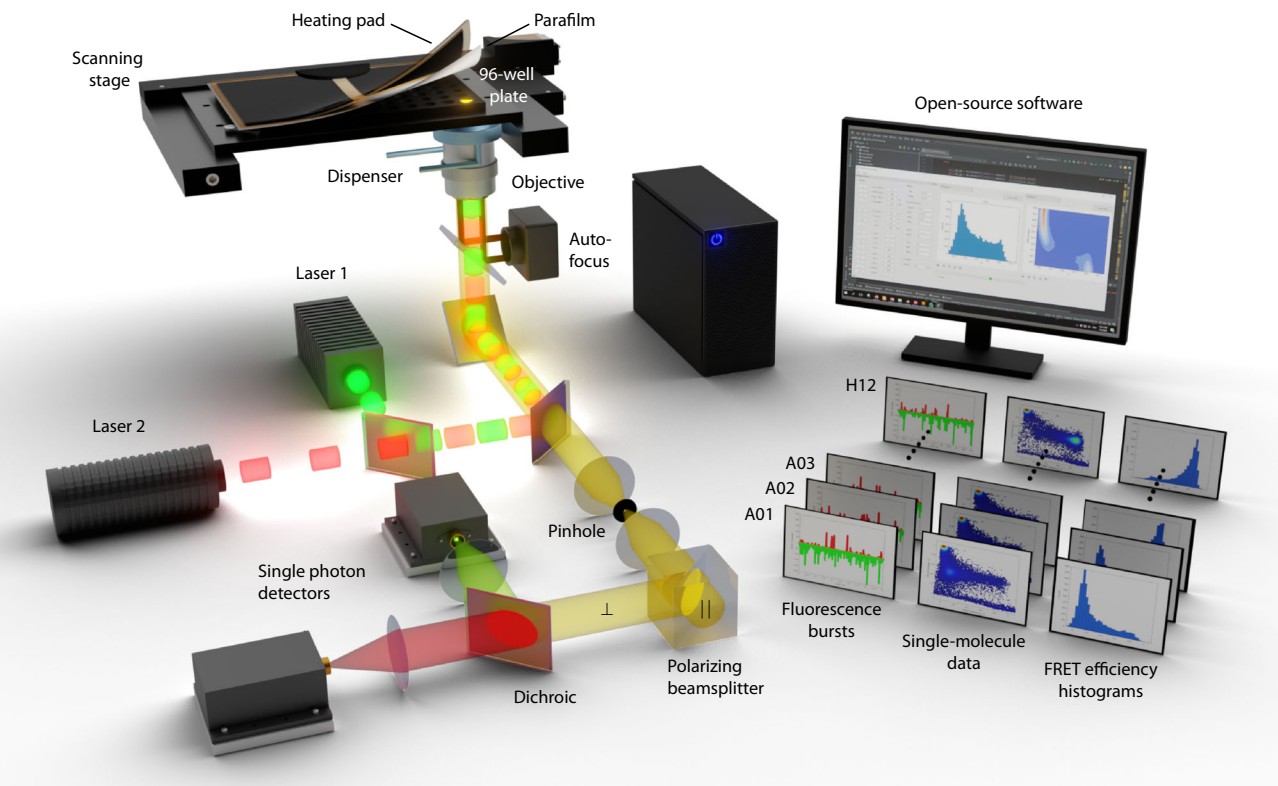

**Fig. 1 | Automated multiwell plate smFRET measurements.** Illustration of the automated single-molecule detection setup implementing a multiwell plate functionality for smFRET experiments. A motorized scanning stage holds the sealed multiwell plate and proceeds from well to well. A liquid dispenser replaces the immersion medium of the water objective and the autofocus maintains the objective focus at a fixed position in solution. Single-molecule fluorescence recordings are performed using a multiparameter confocal fluorescence microscope equipped with pulsed lasers, a high-numerical water objective, and spectrally- and polarization-sensitive single-photon detectors (for simplicity only one polarization is illustrated; see also Supplementary Information). An integrated open-source software suite (available on Github) with graphical user interfaces performs data acquisition, processing, analysis, and visualization. Data can be visualized for each well or ranges of wells (A01, A02, …, H12).

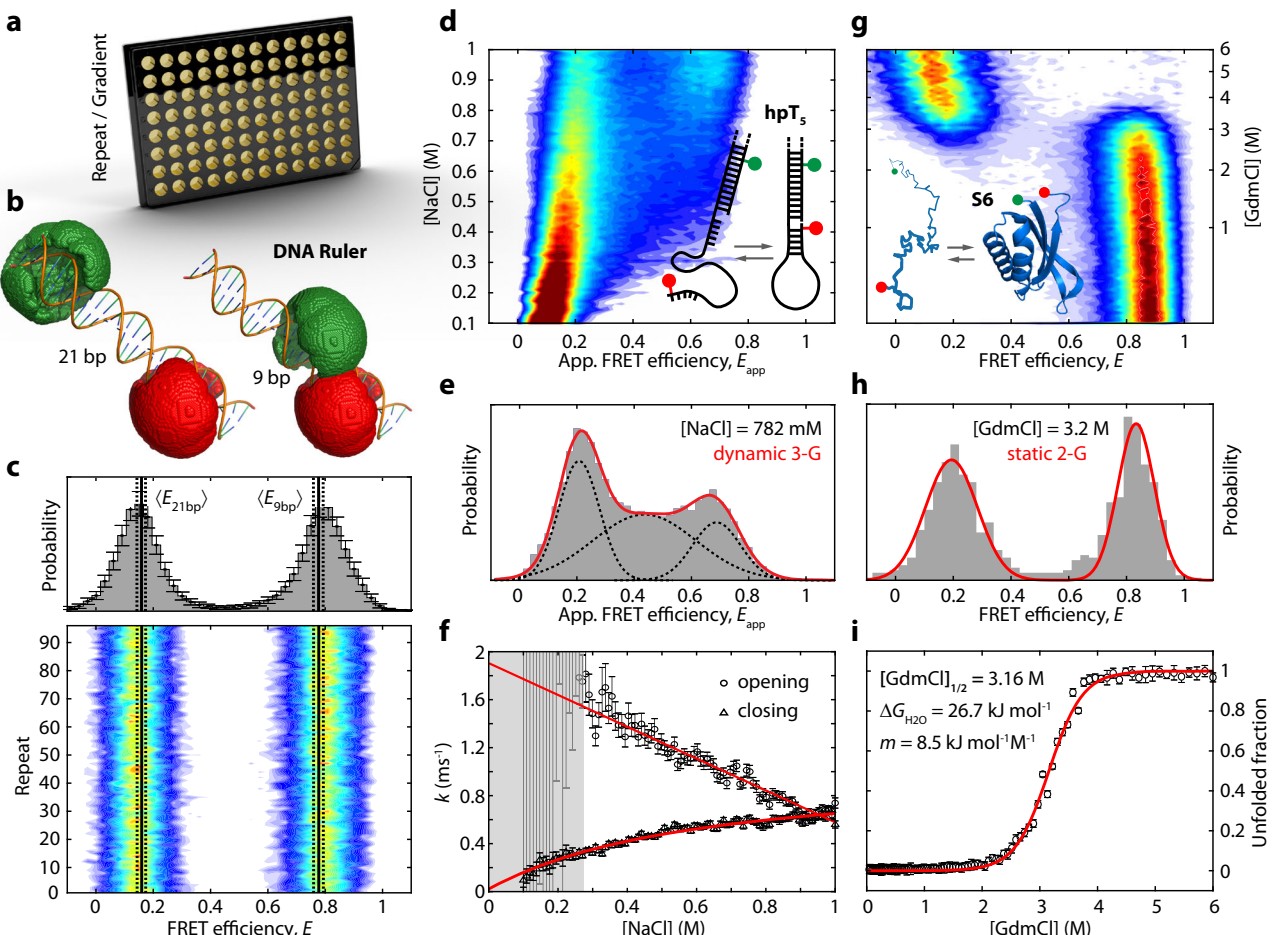

**Fig. 2 | Probing conformational changes of biomolecules by multiwell plate smFRET. a** The multiwell plate format provides the convenience to screen multiple sample repeats or concentration gradients. **b** Evaluation of the accuracy and precision of multiwell plate smFRET measurements using rigid DNA ruler constructs. 96 independent, but identical repeats of a mixture of two double-stranded DNA ruler constructs with 21- and 9-bp spacing between the acceptor (red) and donor (green) fluorophores. **c** Cumulated FRET efficiency histogram of all 96 wells (top) and the two-dimensional (2-D) histogram of $E_{FRET}$ versus multiwell plate repeat (bottom). The error bars indicate the standard deviation (SD) derived from the 96 measurement repeats. Expected $E_{FRET}$ for the two constructs (calculated by accessible volume (AV) simulations[38]), are indicated as black solid lines. The confidence intervals (black dashed lines) are derived from the uncertainty of the Förster radius. **d** 2-D $E_{FRET}$ histogram of the salt-dependent structural dynamics of the hpT5 DNA hairpin in a 96-well plate. Inset: Schematic of the hairpin structure with donor and acceptor fluorophore positions indicated. **e** $E_{FRET}$ histogram of hpT5 at

782 mM NaCl fitted with a dynamic 3-Gaussian model[5,40,41] (red line and black dashed lines). The intermediate $E_{FRET}$ population originates from molecules changing their conformation between open and closed state during the diffusion through the confocal volume. **f** Opening rates of hpT5 decrease linearly with increasing NaCl concentration (red straight line). Closing rates of hpT5 exhibit a non-linear behavior at lower salt concentrations, which is well described by a model considering the apparent concentration (red curved line). Data are presented as predicted value +/−68% confidence interval (CI) as derived by the dynamic 3-G fit. **g** 2-D $E_{FRET}$ histogram of GdmCl-induced unfolding of the protein S6 in a 96-well plate format. Inset: Schematic of the protein with donor and acceptor fluorophore positions indicated. **h** FRET efficiency histogram of S6 at 3.2 M GdmCl (gray bars) and the double Gaussian fit (red line) to quantify the fraction of unfolded molecules. **i** Fraction of unfolded S6 molecules as a function of GdmCl concentration. Data are presented as predicted value +/−68% CI as derived by the static 2-G fit. Data was fitted using Eq. 11 (red line). Source data are provided as a Source data file.

temperature of 25 °C at high salt concentrations, thereby rapidly interconverting between the open and closed conformations on the millisecond timescale, as previously reported[39]. To explore the salt-dependent opening and closing rates of hpT5, we designed a 96-step gradient of NaCl from 0.1 M to 1 M and added ~100 pM of the DNA hairpin to each well. At low salt concentrations, the hairpin appeared mostly in the open, low FRET state ($E_{app,O} ≈ 0.1$). Upon increasing salt concentrations, we observed the appearance of a high FRET population ($E_{app,C} ≈ 0.7$) representing the fully annealed DNA hairpin structure. In addition to the open and closed state populations at low and high FRET efficiencies, the hairpin also exhibits a population at intermediate FRET efficiencies, as shown for $[NaCl] = 782$ mM (Fig. 2e and Supplementary Fig. 5). FRET fluctuation analysis of individual fluorescence bursts (Supplementary Methods and Supplementary Fig. 5) revealed that this population, termed bridge population, originates

from hairpin molecules exhibiting dynamic interconversion dynamics between the open or closed conformation during the ~1-ms-long passage time through the confocal observation volume (i.e., millisecond dynamics). In order to quantify the interconversion rates, $k_{open}$ and $k_{close}$, between the open and the closed states of hpT5, respectively, we used a dynamic 3-Gaussian (3-G) approximation as described earlier[5,40,41]. Using this approach, we modeled the $E$-histogram for each NaCl concentration with three coupled Gaussian distributions corresponding to the open, bridge-like, and closed population (Fig. 2e, solid and dashed lines, Supplementary Fig. 5B). The extracted opening and closing rates for the DNA hairpin are plotted in Fig. 2f. We observed a linear salt dependence of the opening rate (red straight line) with a decline of $m_{open} = −(1.31 ± 0.04)$ ms⁻¹M⁻¹ and an extrapolated rate at zero salt of $k^0_{open} = (1.89 ± 0.03)$ ms⁻¹. By contrast, the closing rate deviates from a linear behavior and exhibits an unexpected curvature

at lower salt concentrations towards smaller rates. To explain this behavior, we developed a model to account for this curvature by considering the apparent local concentration $[A]$ of the 5-bp strand around its complementary strand $\bar{A}$ (i.e., proximal stem of hpT$_5$ connected by the 21-nt long loop). Briefly, the concentration $[A]$ was calculated from the spherical volume that is spanned by the average distance $R_{A-\bar{A}}$ between the two ends of the 21-nt long loop, where $R_{A-\bar{A}}$ was derived from the root-mean-square end-to-end distance of a worm-like chain polymer with contour length $l_c = 14.2$ nm and a calculated ionic-strength dependent persistence length $l_p(I)$[42] (Eq. 9, Supplementary Information). Strikingly, the salt-dependent increase of the apparent concentration $[A]$, as modeled by this approach, reproduces the observed curvature of the closing rate $k_{\text{close}}([\text{NaCl}])$, as shown in Fig. 2f (curved red line). We find an apparent closing rate of $k'_{\text{close}} = (0.305 \pm 0.007) \times 10^6 \, \text{s}^{-1} \, \text{M}^{-1}$, which is in good agreement with earlier reports[42]. Furthermore, from the lower and upper limit of the persistence length of the loop ($l_p(\infty) = 0.75$ nm and $l_p(0) = 2.09$ nm at infinite and zero concentration of NaCl, respectively) a lower and upper boundary of the hpT$_5$ closing rate can be predicted as $0.022 \, \text{ms}^{-1} < k_{\text{close}} < 1.02 \, \text{ms}^{-1}$. Taken together, our multiwell plate experiment on the DNA hairpin yielded detailed insights into the molecular kinetics of this dynamic nucleic-acid system, which would be challenging to extract in independent single-well measurements. Especially the curvature in the low-salt regime could be easily missed at reduced sampling density. In fact, we estimate, with bootstrapping, that the variability (i.e., accuracy) of opening rates decreases with the number of probed conditions. We found that the standard deviation drops from $\sigma_5 = 0.154 \, \text{ms}^{-1}$ for 5 measured samples to $\sigma_{70} = 0.011 \, \text{ms}^{-1}$ for 70 measured samples, demonstrating that a fine-sampled screen with 70 different conditions decreases the standard deviation by 14-fold.

Besides kinetics, many biophysical studies are interested in thermodynamic stabilities of proteins or protein complexes. One possibility to assess thermodynamic stability of proteins is to titrate the protein with a destabilizing chaotropic reagent or a denaturant in small increments and by fitting the data with a transition function (e.g., linear extrapolation method, LEM) to extract the denaturant concentration of half occupancy $[\text{GdmCl}]_{1/2}$, the transition slope $m$, and the change in free Gibbs energy between the folded and unfolded conformation at zero denaturant concentration $\Delta G_{\text{H}_2\text{O}} = m \cdot [\text{GdmCl}]_{1/2}$[43–45]. Such measurements benefit from a high sampling density to avoid over-parameterization and to increase fitting accuracy and precision.

Here, we demonstrate the power of a dense 96-well plate sampling by equilibrium unfolding of the small globular protein S6 with guanidinium chloride (GdmCl). To this extent, we prepared S6 protein site-specifically labeled with a donor and acceptor fluorophore and subjected the protein to increasing GdmCl concentrations in logarithmic steps from 0 M to 6 M, while maintaining the concentration of S6 at ~100 pM. We performed multiwell plate smFRET measurements and probed equilibrium unfolding of S6 in a total of 96 steps by probing each condition for 20 min. We plotted the obtained individual $E$-histograms versus GdmCl concentrations in a 2D histogram (Fig. 2g). This denaturation map shows a compact folded conformation of S6 at $E_F \approx 0.9$ at low GdmCl concentrations and the denaturant-induced unfolding of the protein into an expanded conformation at $E_U \approx 0.2$ beyond 2.5 M GdmCl. Unlike in the case of the DNA hairpin, we did not observe an intermediate FRET population, indicating that S6 does not show millisecond transition kinetics, in agreement with earlier reports[46,47]. Fitting each individual FRET efficiency histogram with a double Gaussian function (2-G) (Fig. 2h), we extracted the average FRET efficiencies as well as the fraction of unfolded molecules at increasing GdmCl concentrations (Fig. 2i). We extracted a transition midpoint of $[\text{GdmCl}]_{1/2} = (3.16 \pm 0.01)$ M and a change in Gibbs free energy of $\Delta G_{\text{H}_2\text{O}} = (26.7 \pm 0.7) \, \text{kJ mol}^{-1}$, in good agreement with earlier reports[47]. Interestingly, the obtained transition slope of $m =$ $(8.5 \pm 0.2) \, \text{kJ mol}^{-1} \, \text{M}^{-1}$ is slightly higher than the reported value of $(4.0 \pm 0.4) \, \text{kJ mol}^{-1} \text{M}^{-1}$, likely originating from the higher pH used in our study (pH 8) as compared to previous studies (pH 6.25)[47,48]. Noteworthy, the high sampling density decreased the standard deviation of the midpoint, $m$-value and thermodynamic stability by 2- to 4-fold (Supplementary Fig. 6D).

In addition to stability, smFRET can also provide structural insights into polypeptide chain properties such as the radius of gyration of the $\Theta$-state (ideal chain), where interactions with the solvent compensate the effect of the excluded volume and the polymer transitions from a globular to a coiled conformation[49]. To demonstrate this on the smFRET data retrieved from S6 unfolding, we extracted the radius of gyration (Supplementary Fig. 6E) of the unfolded peptide chain of S6 by fitting the Sanchez model to the mean FRET efficiencies of the unfolded state[49]. At a scaling exponent of $\nu = 1/2$ (Supplementary Fig. 6F), we found the radius of gyration of the $\Theta$-state to be $R_{G,\Theta} = (2.38 \pm 0.17)$ nm, remarkably close to the theoretical prediction of 2.21 nm (Supplementary Methods). Interestingly, the compaction factor $\alpha = R_G/R_{G,\Theta}$ (Supplementary Fig. 6G) of our structural analysis reveals an early coil-to-globule transition of S6 at a GdmCl activity below the actual folding transition at $a_{\text{GdmCl}} = 1.73$, as also observed for the cold shock protein CspTm and spectrin domain R17[49].

Taken together, our automated multiwell plate smFRET platform allowed us to explore kinetic and thermodynamic parameters governing biomolecular folding in smFRET experiments at very high resolution. The consistency between the kinetic rates and thermodynamic stability of the DNA hairpin and the protein S6 with previous reports demonstrate the reliability of the multiwell plate approach. Moreover, we have discovered insights into biomolecular folding mechanisms including the non-linear closing dynamics of the DNA hairpin at low salt concentrations and the early coil-to-globule transition of the protein S6. Given the breadth and depth of information that can be gained from a single multiwell plate smFRET measurement, we anticipate that the acquisition of high-content smFRET datasets using this format will open new possibilities for discovery in biomolecular folding and dynamic structural biology.

## Observing binding modes of multiple proteins to a single substrate by multiwell plate smFRET

The previous examples demonstrated the ability of our platform to precisely sample changes of molecular conformations and kinetics upon altering solution conditions. Another opportunity by a multiwell plate smFRET assay is to explore target binding modes of multiple, competing reaction partners. A prominent example is the competitive DNA binding of the single-stranded DNA binding protein SSB and the DNA strand-exchange protein RecA. Both proteins readily bind to single-stranded DNA (ssDNA), however, their interaction mode is different. SSB occludes 35- or 65-nt-long stretches on ssDNA and dissolves DNA secondary structures[50]. RecA, by contrast, is known to form a filament on ssDNA with a 3-nt footprint, and is a key player in homologous recombination[51]. Single-molecule experiments have discovered already a direct interaction and competition of RecA and SSB, with RecA nucleation being facilitated by SSB, likely by RecA–SSB complexes[52–54]. However, this intricate interactive behavior and likely multiple pathways of binding make it difficult to explore the full parameter space of affinities and nucleation by sets of single-well measurements. The multiwell plate format offers the possibility to apply concentration gradients of two molecules against each other (Fig. 3a), making it easier to identify competitive and cooperative effects as well as to pinpoint a reaction scheme and to extract dissociation constants. To demonstrate this capability, we performed multiwell plate smFRET measurements of SSB and RecA and studied interactive binding of both proteins to ssDNA.

We designed a DNA construct with an 18-bp double-stranded DNA region carrying an acceptor fluorophore and a 70-nt long thymidine

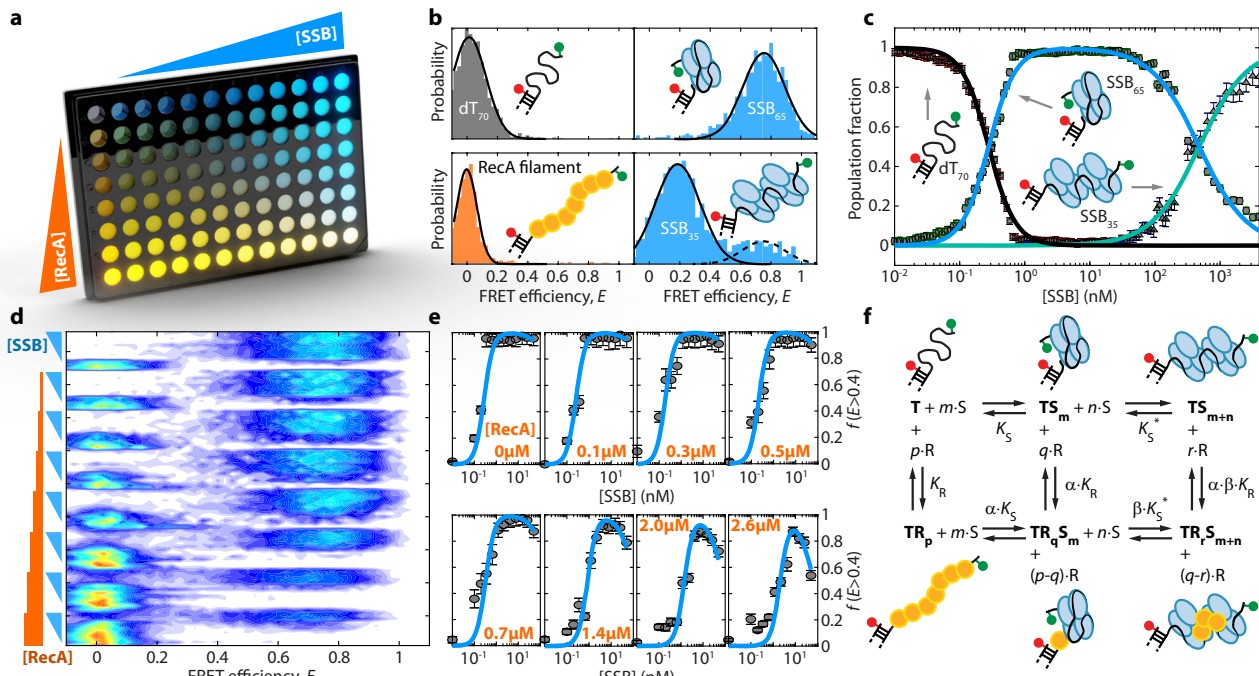

**Fig. 3 | Observing binding modes of multiple proteins to a single substrate by multiwell plate smFRET. a** Competitive binding of RecA and SSB to ssDNA probed in a 96-well plate measurement by a combined variation of the concentration of SSB (0 to 50 nM) and RecA (0 to 2.6 µM) using the DNA construct $dT_{70}$ as a substrate. **b** $E_{FRET}$ histograms of $dT_{70}$ in absence of proteins, presence of SSB at low and high concentration and RecA. The construct allows to identify unique $E_{FRET}$ for unbound DNA, $SSB_{65}$ and $SSB_{35}$ binding modes, as well as a RecA filament on ssDNA. **c** Speciation curves as obtained from a multiwell plate smFRET experiment of $dT_{70}$ subjected to increasing concentration of SSB ranging from 0 to 4 µM. Data are presented as predicted value +/−68% CI as derived by the Gaussian fit. **d** 2-D $E_{FRET}$ histogram of the competitive binding of RecA and SSB to $dT_{70}$ as obtained from a 96-well plate smFRET measurement. Bars and wedges on the left side depict RecA and SSB concentrations, respectively. Increasing RecA concentrations shift the transition to the $SSB_{65}$ binding to higher concentrations of SSB. **e** Fraction of molecules with FRET efficiency $E > 0.4$ versus [SSB] for increasing RecA concentrations. Fractions were fitted with an optimized 6-state equilibrium model (blue lines). Data are presented as mean +/− SD. The SD is derived from the counting uncertainty by simple error propagation. **f** 6-state model used to describe competitive binding of SSB and RecA to $dT_{70}$. T: empty $dT_{70}$; $TS_m$: SSB on $dT_{70}$ in $SSB_{65}$ binding mode; $TS_{m+n}$: SSB on $dT_{70}$ in $SSB_{35}$ binding mode; $TR_p$: RecA filament on $dT_{70}$; $TR_qS_m$: mixed state of $SSB_{65}$ and RecA on $dT_{70}$; and $TR_rS_{m+n}$: mixed state of $SSB_{35}$ and RecA on $dT_{70}$. The values $K_S$, $K_S^*$, $K_R$, $\alpha \cdot K_R$, $\alpha \cdot \beta \cdot K_R$, $\alpha \cdot K_S$, $\beta \cdot K_S^*$ denote the apparent dissociation constants with the corresponding Hill coefficients $m$, $n$, $p$, $q$, and $r$. The RecA concentrations at half occupation ($C_{i,1/2} = \sqrt[i]{K}$) was extracted from the fit in panel E (blue lines). Values are given in the main text. Source data are provided as a Source data file.

($dT_{70}$) ssDNA overhang on the 3'-end terminated by a donor fluorophore (Fig. 3b inset, Supplementary Table 4). The proximity of the donor and acceptor fluorophores allows monitoring the compaction of the $dT_{70}$ overhang upon binding of SSB or RecA (Fig. 3b). In absence of any protein, the ssDNA exhibits a low FRET efficiency ($E_{dT70} \approx 0.05$) as the DNA is only weakly collapsed. In the presence of low concentrations of SSB (e.g., 1 nM), SSB binds to the ssDNA in the $SSB_{65}$ binding mode, where 65 nts are occluded by SSB, leading to a FRET efficiency of $E_{65} \approx 0.8$ (Fig. 3b). With increasing SSB concentrations (>50 nM), the $SSB_{65}$ binding mode transitions into the $SSB_{35}$ binding mode, incorporating two tetramers of SSB (Fig. 3b) and, hence, leading to a more expanded $dT_{70}$ conformation with a FRET efficiency $E_{35} \approx 0.2$. RecA, in presence of ATP, forms a nucleoprotein filament, thereby stretching ssDNA. Hence, binding of RecA to $dT_{70}$ results in its elongation beyond the dynamic range of FRET, yielding a narrow distribution of FRET efficiencies of $E_{RecA} \approx 0$. Notably, the signature of RecA binding is clearly distinguishable from the two major binding modes of SSB (Fig. 3b)[55]. In summary, the high FRET contrast of the three states, $SSB_{65}$, $SSB_{35}$, and RecA filament, allows shedding light on the interactive behavior of both proteins in the presence of ssDNA.

In a first experiment, we studied SSB binding to ssDNA alone. To this end, we performed a multiwell plate measurement of $dT_{70}$ (~100 pM) subjected to increasing SSB concentration ranging from 0 to 4 µM (Supplementary Fig. 7). We determined the fractions of molecules in the $dT_{70}$ (rectangle), $SSB_{65}$ (circle) and $SSB_{35}$ (triangle) state as a function of SSB concentration (Fig. 3c). The fractions were then modeled by Eq. 15 (Supplementary Information) to derive the concentrations at half occupancy of $c_{S,1/2} = (278 \pm 1)$ pM and $c_{S,1/2}^* = (480 \pm 17)$ nM and the corresponding Hill coefficients of $m = (2.12 \pm 0.07)$ and $n = (1.19 \pm 0.04)$ for the $SSB_{65}$ and $SSB_{35}$ binding mode, respectively. The rapid continuous mapping across different concentrations agrees well with the simple binding theory (Eq. 15) and earlier reports[50,52,56].

Subsequently, we performed a multiwell plate measurement of $dT_{70}$ with 96 different combinations of [RecA] and [SSB] to study the competition between RecA filament formation and $SSB_{65}$ binding. To this end, we varied the SSB concentration in 12 steps from 0 to 50 nM along the columns of the plate and the RecA concentration in 8 steps from 0 to 2.6 µM along the rows of the plate (Fig. 3a). In the measurements without RecA (top row, Fig. 3d), the transition from the broad $dT_{70}$ state ($E_{dT70} \approx 0.05$) to the $SSB_{65}$ binding mode ($E_{65} \approx 0.8$) appears at a low concentration between 0.2 and 0.35 nM agreeing well with our previously determined $c_{S,1/2} = 0.28$ nM, and the independent observation of the change in fluorescence anisotropy of the acceptor (Supplementary Fig. 8C). With increasing RecA concentrations, the occupation of $SSB_{65}$ shifted to higher SSB concentrations. At the same time, the RecA population at $E_{RecA} \approx 0$ became more abundant, reflecting a modulation of the apparent dissociation constant of SSB by competitive binding of RecA. Interestingly, at high RecA concentrations ([RecA] = 2 − 2.6 µM) a shift of the high FRET population to lower FRET efficiencies is observed, which suggests an unknown state of combined RecA and SSB binding.

For a quantitative analysis of the interactive binding of SSB and RecA to ssDNA, we extracted the fractions of DNA molecules bound in the $SSB_{65}$ mode $f(E > 0.4)$ from the 2D histogram (Fig. 3e). As qualitatively observed, at higher RecA concentrations, the transition to the SSB binding mode occurred at higher SSB concentrations. Surprisingly, we observed a drop of the maximal fraction of $SSB_{65}$ at [RecA] > 0.5 µM and [SSB] > 10 nM. The depopulation of the $SSB_{65}$ state at elevated [RecA] supports the presence of mixed RecA–SSB states. Considering the two SSB binding modes and the presence of a RecA-filament, we build a 6-state model (Fig. 3f), which contains the known $dT_{70}$ (T), RecA filament ($TR_p$), $SSB_{65}$ ($TS_m$) and $SSB_{35}$ ($TS_{m+n}$) states, as well as the two mixed states of RecA-SSB ($TR_qS_m$) and RecA-2xSSB ($TR_rS_{m+n}$). Here, the values $K_S$, $K_S^*$, $K_R$, $\alpha \cdot K_R$, $\alpha \cdot \beta \cdot K_R$, $\alpha \cdot K_S$, $\beta \cdot K_S^*$ denote the respective dissociation constants with the corresponding Hill coefficients $m$, $n$, $p$, $q$, and $r$. The 6-state reaction scheme allowed us to model the fractions of molecules with $E > 0.4$ (Fig. 3e). Since it was unclear where the states $TR_qS_m$ and $TR_rS_{m+n}$ appear on the FRET axis, and thus what states contributed to $f(E > 0.4)$, we performed 75 different fittings with varying state and probability configurations using Eq. 16 (Supplementary Information) and the previously determined values $c_{S,1/2}$, $c_{S,1/2}^*$, $m$ and $n$. For each combination of state and probability configurations, the reduced chi-squared ($\chi_r^2$) was calculated from the residual, taking the number of fitting parameters into account. We found the smallest $\chi_r^2$ value ($\chi_r^2 = 0.108$) for the full reaction scheme involving all 6 states, where $f(E > 0.4)$ describes the combined fraction of state $TS_m$ ($SSB_{65}$) and $TR_qS_m$ (RecA-SSB) (Supplementary Fig. 10), an SSB–RecA complex. Looking at the formation of a RecA-filament, we find half concentrations of occupancy of $c_{Rp,1/2} = (425 \pm 91)$ nM, and for RecA-SSB formation a value of $c_{Rq,1/2} = (237 \pm 76)$ nM, and for RecA-2xSSB a value of $c_{Rr,1/2} = (278 \pm 94)$ nM with the corresponding Hill coefficients of $p = (4.9 \pm 1.5)$, $q = (1.9 \pm 1.5)$ and $r = (3.1 \pm 1.6)$. Hence, our data revealed that RecA affinity is increased 1.8-fold by the presence of a $SSB_{65}$-complexed ssDNA and 1.5-fold by a $SSB_{35}$-complexed DNA. SSB facilitating RecA filament formation was previously observed[52–54], yet, it was impossible to quantify this enhancement.

Taken together, our measurements illustrate the possibility to screen three or more component systems for cooperativity or competition within a single 96-well plate smFRET experiment and, by extension, reveal new, unexpected cooperativity and competition effects.

## Multiwell plate smFRET screening of drug–protein interactions

smFRET experiments are increasingly employed to study the molecular mechanisms of small-molecule binding to target proteins in a variety of applications, ranging from enzyme–ligand interactions to probing the reversal effect of small molecule corrector compounds on protein misfolding[57–61]. However, larger-scale screenings of molecular compounds by smFRET, as used in pharmacological research and drug discovery, have been limited because tools to conduct such time- and labor-intensive measurements are lacking. For example, recently, we used smFRET to study the misfolding and drug rescue mechanism of the cystic fibrosis transmembrane conductance regulator (CFTR), an ion channel protein that is defective in people with cystic fibrosis (pwCF). We used a minimal hairpin model derived from the CFTR transmembrane helices 3 and 4 (TM3/4) carrying a patient-derived mutation and found that misfolding induced by the point-mutation V232D in TM3/4 could be rescued with the drug Lumacaftor[62]. Such experiments required tens of single chamber smFRET measurements with extensive cleaning steps, long equilibration periods, and repeated sample reconstitution for single conditions. Extension of such measurements to larger-scale screenings would benefit massively from an automated multiwell plate format in order to probe multiple small molecules or multiple patient-derived mutations.

Here, we explored such an automation for molecular screening of drug–protein interactions by our multiwell plate smFRET assay. The screen comprised four protein variants and two small molecule compounds. The protein variants consisted of wildtype TM3/4 (WT) and three mutant variants E217G, Q220R, and V232D TM3/4 (Fig. 4a, b and Supplementary Table 4). All three mutations are CF-phenotypic and known to cause maturation defects and misfolding of CFTR[63,64]. The drug molecules were two CFTR interacting correctors, Lumacaftor (VX-809) and Galicaftor (ABBV-2222)[65]. Lumacaftor as well as Galicaftor have been developed to rescue the most common misfolding mutation ΔF508 in CFTR[66,67], but also showed improved maturation of CFTR with V232D[63,68]. To read out misfolding and drug rescue of the transmembrane helices, we attached donor and acceptor fluorophores close to the N- and C-termini of our TM3/4 hairpins and reconstituted the TM3/4 hairpins in lipid vesicles (Fig. 4b). After reconstitution and transfer into a 96-well plate, we read out the degree of misfolding and partial insertion by collecting FRET efficiency histograms in presence of increasing concentrations of corrector compounds (over each row of the 96-well plate). In this context, a high FRET efficiency is related to correct insertion and correct folding, while a lower FRET efficiency indicates partial insertion and misfolding[62]. Hence, we can detect misfolding, and by performing concentration screenings of corrector compounds, we can detect the degree of drug rescue of the hairpin structures by reading out FRET efficiencies and determine an $EC_{50}$ of the drug–protein interaction from dose–response curves.

Using our multiwell plate assay, we collected FRET efficiency histograms for our four hairpin variants and two corrector molecules (Fig. 4c and Supplementary Fig. 11). The histogram for WT TM3/4 showed, in the absence of any corrector, a compact structure with a high fraction of molecules being in the correctly folded, high FRET efficiency state (Fig. 4d, blue dots). Upon addition of either Lumacaftor or Galicaftor, little change of the degree of folding was observed. Also the variants E217G and Q220R TM3/4 mostly retained a closed conformation with high FRET efficiency similar to WT TM3/4, in agreement with our earlier study[69]. V232D TM3/4, on the other hand, in the absence of any corrector, appeared misfolded and adopted a mostly open conformation with low FRET efficiency (Fig. 4c, d). Titration of Lumacaftor restored folding of V232D TM3/4 to a large degree with $EC_{50} = (337 \pm 39)$ µM, which is in excellent agreement to our previously reported $EC_{50}$ of 347 µM (Fig. 4d). Lumacaftor, however, showed no drastic effect on the loop mutants E217G and Q220R TM3/4 as they remained largely folded even in the absence of the corrector, and we observed only a slight stabilization at high concentrations. In the case of Galicaftor, which we did not study previously, we observed that V232D TM3/4 could be rescued from misfolding as well, while the other variants were little affected by increasing Galicaftor concentrations (Fig. 4c, d). Noteworthy, at [Galicaftor] > 400 µM, we detected a deviation of the fluorescence lifetime of the donor and acceptor, which interfered with the detection and, thus, was not considered in our analysis (Fig. 4d, gray area). Interestingly, the observed $EC_{50}$ for the rescue of V232D TM3/4 was $EC_{50} = (80 \pm 24)$ µM and, thus, much lower than for Lumacaftor. Recent in vivo experiments also observed that Galicaftor is 12-fold more potent than Lumacaftor in rescuing full-length V232D CFTR at the plasma membrane. This corroborates that the structural readout of smFRET experiments on TM3/4 misfolding and its rescue by small molecule correctors provides insights into drug-action mechanisms. While CFTR TM3/4 shows a relatively large conformational change upon drug binding, employing multiple, complementary fluorescence parameters, as reported earlier[70], including anisotropy, diffusion time, and fluorescence lifetime, can help identifying molecular interactions with rather small conformational changes.

In summary, our results on TM3/4 hairpin screening illustrate that within a single, automated multiwell plate smFRET experiment, we were able to recover structural information of misfolding events and

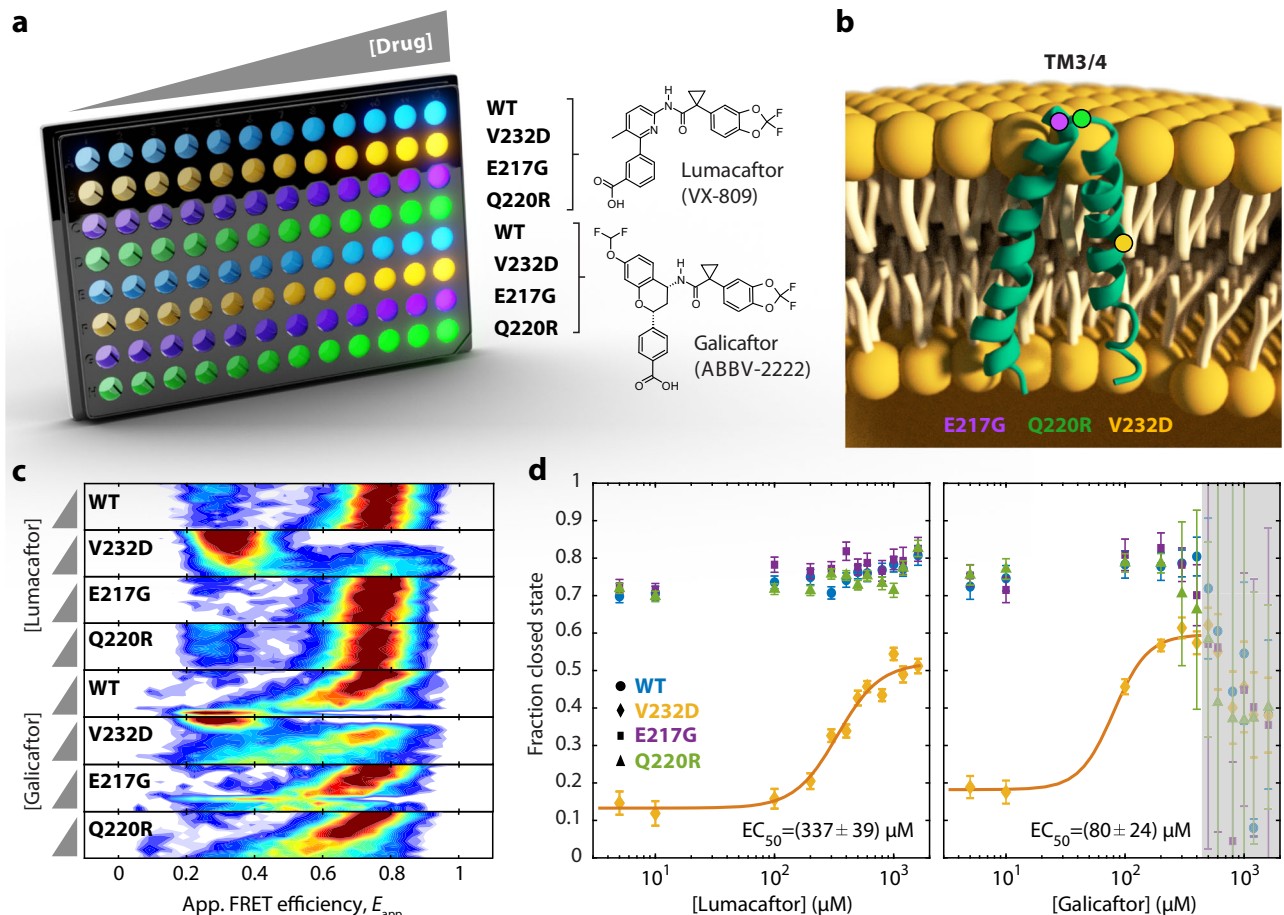

**Fig. 4 | Multiwell plate smFRET screening of drug–protein interactions.**
**a** Titration of reconstituted CFTR TM3/4 hairpins (WT, V232D, E217G, Q220R) with Lumacaftor (VX-809) and Galicaftor (ABBV-2222) over 12 different drug concentrations in a 96-well plate. **b** Schematic of the TM3/4 hairpin structure (PDB ID: 5UAK) in a lipid bilayer. The positions of the transmembrane mutant V232D and of the two loop mutations E217G and Q220R are indicated by colored circles. **c** 2-D $E_{FRET}$ histogram versus drug concentration for WT, V232D, E217G and Q220R TM3/4 in the presence of either Lumacaftor or Galicaftor. The concentration of the drugs

was increased in 12 steps from 5 to 1600 μM (gray wedges). **d** Dose–response curves for TM3/4 variants in the presence of Lumacaftor (left panel) or Galicaftor (right panel). Data are presented as predicted value +/−68% CI as derived by the Gaussian fit. Dose–response curves were fitted with a Hill-type function to yield $EC_{50}$ values for Lumacaftor and Galicaftor. In the case of Galicaftor, data points above 500 μM were excluded from the fit, due to deviation of the fluorescence lifetime of the donor and acceptor (indicated in gray). Source data are provided as a Source data file.

their rescue, illustrating the suitability of our assay for drug screenings. We anticipate that such a multiwell plate assay opens up avenues to use smFRET for the characterization of patient-derived mutations on conformational dynamics and their rescue, thus providing long-sought-after approaches for rational drug design and drug discovery.

## Discussion

Here, we introduced a platform for automated smFRET experiments in a multiwell plate format. With different examples, we illustrated how accurate and precise FRET efficiencies as well as conformational dynamics, molecular competitions, and information on small-molecule–protein interactions can be obtained from multiwell plate measurements. Along with a detailed description of the hardware components, which are all commercially available and can be installed without expert knowledge, we provide an open-source software suite for data acquisition, analysis, and visualization (see Supplementary Methods), offering an easily adaptable approach for other labs to integrate the multiwell plate smFRET assay into their workflows.

All our examples were performed on 96-well plates with a 20-min measurement time per well. This accumulates to a total measurement time of 32 h and was sufficient to collect extensive data for all provided examples. However, our assay can be easily adjusted to either smaller

(e.g., 48-wells) or larger (384-wells) multiwell plate formats, as desirable for the application. Further, the data acquisition software allows to select the specific wells to be probed on the multiwell plate, such that only a subset of conditions can be probed, and in the case of varying measurement statistics, the data acquisition time can be adjusted flexibly. With respect to measurement time, we noted that 32 h measurements can cause loss of fluorescent molecules (e.g., by non-specific adsorption to the 96-well plate). However, such loss can be prevented by the addition of a small percentage of surfactant (e.g., Tween20) or a brief preincubation with BSA to achieve surface passivation.

For confocal smFRET experiments, the measurement of a single condition typically takes about 20 min to 2 h, depending on the system under scrutiny, and is usually followed by chamber cleaning, sample preparation, sample loading and data management, which typically take about 30 min per condition. These laborious and time-consuming steps can be drastically reduced with our multiwell approach. Automatization of data acquisition overcomes repetitive workflows, and multiwell plate handling enables swift sample preparation without repeated preparation steps. Of note, all of the samples described in this work were manually pipetted in an optimized pipetting scheme taking ~2–3 h to prepare an entire plate. These sample preparation

steps can be further improved. Commercial implementations of micro-dispensers allow to fill a 96-well plate within 15 min in an automated fashion (Supplementary Information). Such a rapid, reliable preparation of 96-well plates provides an important step towards high-content screenings by smFRET. In fact, the ease of setting up such multiwell plate experiments will unleash the unique possibility to extensively, yet swiftly, bridge the gap between structural and functional aspects of biomolecular systems in dynamic structural biology and biophysics, and beyond. Of note, the multiwell platform is not limited to 96-well plates but can be readily adjusted to other plate specifications in the open-source code.

Direct extraction of kinetic information from diffusion-based confocal smFRET experiments is typically limited to the millisecond timescale due to the short observation time (i.e., ~1 ms). However, several techniques have been developed to overcome this limitation. These include, for example, the usage of bigger pinholes to expand the confocal volume and thus enable longer observation times, analysis routines that can extract kinetics on longer timescales such as Recurrence Analysis of Single Particles (RASP)[6], or vesicle encapsulation to slow down diffusion. These techniques can be also applied to our automated approach, thereby extending the extraction of kinetics beyond the millisecond timescale. In fact, in the case of slow equilibration dynamics on the timescale of minutes to hours, the multiwell approach offers the possibility to use the temporal information from well-to-well changes or enables the return to earlier wells for a repeat of conformational sampling. An intermediate time range of seconds to minutes is typically sampled using immobilized molecules combined with microfluidics, which is an attractive, complementary approach[1,71]. Moreover, our multiwell plate format should also be applicable to fluorescence correlation spectroscopy measurements that enable the extraction of dynamics on the second to nanosecond timescales[3,72,73].

Overall, with our automated multiwell plate platform, we open up unique possibilities to use high-content smFRET data for biomolecular screening and drug discovery. The increased sampling of multiwell plate smFRET, as demonstrated in our work, allows, for instance, to (i) discover unexpected interactions in multi-component systems; (ii) screen many different small molecules for affinities and effects on molecular conformations; and (iii) discover subtle conformational changes, which are typically inaccessible in single-well measurements due to low parameter space sampling. Together with our open-source cross-platform software suite and the easy-to-implement additions to already available smFRET setups, we anticipate that multiwell plate smFRET will enable the community to set up such a system in their labs and to gain in-depth insights into biological systems, spanning from protein folding to nucleic-acid structures and protein–small molecule interactions.

## Methods

### DNA and protein preparation
DNA samples, including fluorescently labeled oligos, were commercially obtained and annealed. Proteins were either purchased commercially or expressed recombinantly and purified. Details are given in the Supplementary Methods (pages S5 to S6 and Supplementary Table 4).

### Multiwell plate experiments
We loaded our 96-well plates in 2 to 3 pipetting cycles using prepared stock solutions of samples and buffers (Supplementary Table 5). Detailed concentrations are given for each well in Supplementary Tables 7 to 11. We describe below briefly the range of conditions.

**DNA ruler experiments.** Annealed DNA rulers were diluted to ~100 pM in buffer (20 mM Tris-HCl, pH 8, 50 mM NaCl) and distributed into each well (Supplementary Table 5).

**DNA hairpin experiments.** Annealed DNA hairpin $hpT_5$ was diluted to ~100 pM in buffer (20 mM Tris-HCl, pH 8) with well-specific NaCl concentrations ranging from 100 to 1000 mM in each well (Supplementary Tables 5 and 7).

**S6 unfolding experiments.** S6 was diluted to ~100 pM in buffer (50 mM Tris-HCl pH 8, 150 mM NaCl, 2 mM TCEP) with well-specific GdmCl concentration ranging from 0 to 6 M (Supplementary Tables 5 and 8).

**RecA and SSB experiments.** DNA $dT_{70}$ was diluted to ~100 pM in buffer (50 mM Tris-acetate pH 7.7, 5 mM Mg-acetate, 50 mM Na-acetate) and SSB (Promega Corporation, USA) was supplemented in specified concentrations (Supplementary Tables 5 and 9). For the SSB–RecA competition, the buffer was supplemented with 16 mM ATP and RecA (New England Biolabs, USA) as well as SSB gradients were added to the respective rows (Supplementary Table 10).

**TM3/4 experiments.** Reconstituted TM3/4 variants in POPC LUVs were diluted to ~100 pM in buffer (50 mM Tris-HCl pH 7.4) and added to each well. Wells were supplied with a gradient of small molecule concentrations ranging from 5 to 1600 μM (Supplementary Tables 5 and 11).

### Automated multiparameter single-molecule detection setup
Experiments were carried out using a single-molecule confocal fluorescence microscope with pulsed-interleaved excitation and fluorescence anisotropy detection as shown in Fig. 1 and described in detail in the Supplementary Methods (pages S3 to S5). Briefly, the microscope was equipped with a motorized sample stage (ASR100B120B, Zaber Technologies, Canada), a heating pad (Lerway, China), and an autofocus system (Perfect Focus System, Nikon, Japan). The immersion water was supplied by a liquid dispenser (Märzhäuser Wetzlar, Germany). The multiwell plates used in this study were glass bottom 96-well plates from IBL Baustoff+Labor GmbH, Austria.

### smFRET data analysis
Data analysis was performed using pyBAT and pyVIZ as well as customized scripts for quantification of center positions, fractions, kinetic rates, and other extracted parameters. Single-molecule events were identified from the acquired photon stream by a burst search algorithm. Details about the primary and secondary data analysis procedures are given in the Supplementary Methods (pages S7 to S9 and individual analysis sections).

### Statistics and reproducibility
No statistical method was used to pre-determine sample size. Exclusions were made solely for data points exhibiting exceptionally large confidence intervals (DNA hairpin kinetic data) or exhibiting impaired donor and acceptor fluorescence (TM3/4 Galicaftor data), enhancing the overall quality of the data analysis. To filter double-labeled and single-labeled molecules we applied a stoichiometry and brightness ratio filter. Filter parameters are provided for each experiment in the Supplementary Methods sections. The experiments were not randomized. The investigators were not blinded to allocation during experiments and outcome assessment.

### Reporting summary
Further information on research design is available in the Nature Portfolio Reporting Summary linked to this article.

## Data availability
Raw datasets measured and analyzed in the current study are available on OpARA: DNA ruler (https://doi.org/10.25532/OPARA-202); DNA

hairpin (https://doi.org/10.25532/OPARA-206); S6 protein (https://doi.org/10.25532/OPARA-209); SSB and RecA competition (https://doi.org/10.25532/OPARA-205); CFTR TM3/4 (https://doi.org/10.25532/OPARA-207). Source data are provided with this paper.

## Code availability

The software used in this manuscript is freely available from the GitHub repository: https://github.com/SchlierfLAB/autoFRET and via https://doi.org/10.5281/zenodo.8269134. The Python-based platform software suite autoFRET comprises three components: Data Acquisition (pyMULTI), Analysis (pyBAT), and Visualization (pyVIZ). A detailed description of the software package can be found in the Supplementary Methods and on GitHub, including an example dataset and instructions on initial analysis steps.

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

## Acknowledgements

We thank all members of the Schlierf lab for lively discussions during the development of this project. This research was funded by TU Dresden core funds (M.S.), the DFG SCHL1896/3-1 (M.S.) and SCHL1896/4-1 (M.S.), the BMBF OptiZeD Grant Z22E511 (M.S.), the European Social Fund and co-financed by tax funds based on the budget approved by the members of the Saxon State Parliament (M.Sche.) and by a grant from Mukoviszidose Institut gGmbH, Bonn, the research and development arm of the German Cystic Fibrosis Association Mukoviszidose e.V. (M.S.). We acknowledge support by the European Research Council (ERC) under the European Union's Horizon 2020 Framework Programme through the Marie Skłodowska-Curie Grant MicroSPARK (agreement no. 841466; G.K.), the Herchel Smith Funds of the University of Cambridge (G.K.), and the Wolfson College Junior Research Fellowship (G.K.).

## Author contributions

M.S., A.H., G.K., and M.Sche. conceptualized the study, A.H., K.S., and P.S. devised and validated the methodology and developed the open source software, A.H., K.S., M.Sche., and N.C. performed the investigation, A.H. and K.S. performed the formal analysis, G.K. and M.Sche. provided resources, A.H. and M.S. supervised the project, A.H., N.C., M.Sche., and M.S. wrote the original draft and all other authors reviewed and edited it. M.S., G.K., and M.Sche. acquired funding for the project.

## Funding

## Competing interests
The authors declare no competing interests.
