## [Peer Review File · Nature Communications]

REVIEWERS' COMMENTS:

Reviewer #1 (Remarks to the Author)

In this submission, Hartmann et al. describe an in-house built confocal smFRET instrument capable of automatically collecting data in 96-well plates. They provide a detailed hardware description and open-source software suite, enabling non-experts to assemble the instrument, collect, visualize, and analyze the data. Additionally, they tested their instrument with four cases: a simple rigid DNA FRET construct, hpT5 structural dynamics, protein binding modes, and drug-protein interactions. It took 32 hours to scan 96 wells, but the system can be adjusted for fewer or more wells. To achieve this performance, the authors developed three major additions to a commercial single-molecule confocal machine: an X-Y stage with < 40-micron accuracy, a heating pad on top of the well to prevent condensation, and a liquid dispenser to replenish objective immersion. Although the methodology is sound and there are enough details to reproduce the instrument, the impact on the field remains unclear due to the numerous commercially available instruments, such as the Nikon A1R HD25, Leica TCS SP8, and Zeiss LSM 880 with Airyscan. Overall, I think this is a rigorous instrumental development, but has limited application beyond their own laboratories.

My detailed comments:

1. What unique features does this in-house built instrument offer that commercial platforms cannot? In fact, Nikon's Ti2-E stage can perform both fine and coarse stepping from nm to cm, outperforming the reported instrument here. Nikon's NIS-elements software also integrates hardware, automation, image processing, and analysis.
2. This platform may be valuable for upgrading existing confocal microscopes without motorized stages. However, fitting an X-Y stage into an existing microscope is case-by-case, and usually, X-Y stages come with their own control software. In this regard, this submission lacks details on how to implement their platform onto a specific model of confocal microscope and appears redundant.
3. The impact of the 32-hour scanning time lapse from the first sample to the last sample on data quality and analysis is not discussed.
4. The limitation of a 1 ms observation window for many slower biological processes is not addressed.
5. For the kinetic information extraction in Figure 2F, it is unclear how the units of k_{on} and k_{off} can be precisely determined to be ms^{-1} . Since single-molecule bursts can only provide the relative distribution of FRET species, as shown in 2E, any relative k_{open} and k_{off} will fit their data for units of any time scale. The reason for precisely pinning the unit to ms^{-1} is not clear.
6. A discussion comparing this in-house built instrument to the commercial instruments mentioned above would improve the next version of this submission.
7. A discussion of how to integrate this new platform into some specific models to guide nonexperts in working on their own models would be helpful.

Reviewer #2 (Remarks to the Author)

The manuscript entitled "Farewell to single-well: An automated single-molecule FRET platform for high-content, multiwell plate screening of biomolecular conformations and dynamics" and authored by Hartmann et al. describes the development of an automated platform for single molecule FRET

measurement in a multiwell plate format. This is a notable technical advancement in smFRET study of conformational fluctuations of nucleic acids and protein dynamics on the timescale around or longer than ms. The high-content capability of the platform would be well suited for biomolecular screening and drug discovery.

It should be appreciated that, to demonstrate the effectiveness and evaluate the improvement in precision and accuracy of smFRET measurement using the new platform, the authors carried out comprehensive work on various well-studied systems ranging from the conformational fluctuation of DNA hairpin to protein-drug interaction. The manuscript are generally well-written with detailed descriptions of the materials and methods in supporting information.

Below are several specific points for the authors to consider in revision.

1. The significance and the list of the applications of high-content fluorescence imaging in ultrasensitive measurement could be expanded in the introduction. Several platforms have been commercialised. For example, a closely related method - fluorescence correlation spectroscopy - can now be measured in a multiwell plate format.

2. Very significant improvement in precision and accuracy in smFRET measurement. But the limitation of the method in terms of the timescale of the conformational kinetics it can measure should be acknowledged. Other related methods which cover different time windows could be discussed.

3. Figure legends are generally too long. They could be made more concise by moving some descriptions to the main text or supporting information.

4. Please quote the standard deviations for " $\langle E_{9bp} \rangle = 0.797$ and $\langle E_{21bp} \rangle = 0.146$ " (in page 7).

5. Was temperature regulated during measurements? It is not clear at what temperature the effects of salt on the kinetics of DNA hairpin was determined (page 8).

6. Y-axis title is missing in Fig 2C. Regarding Fig 2F, I would suggest that the x-axis scale starts from 0 instead of 0.1 so the Y-intercepts could be clearly visualized. Also give the reasoning why opening rates under low salt conditions were not included.

7. A general multiwell plate might not be compatible with confocal imaging. The bottom thickness of the well plate used should be described either in the main text or supporting information.

Reviewer #3 (Remarks to the Author)

In the manuscript titled "Farewell to single-well: An automated single-molecule FRET platform for high-content, multiwell plate screening of biomolecular conformations and dynamics", the authors introduced a platform for automated smFRET experiments in a multiwell plate format. With different examples, they have illustrated how accurate and precise FRET efficiencies, as well as conformational dynamics, molecular competitions, and information on small-molecule-protein interactions, can be obtained from multiwell plate measurements. Along with a detailed description of the hardware components, they have also provided an open-source software suite for data acquisition, analysis, and visualization offering an easily adaptable approach for other labs to integrate the multiwell plate smFRET assay into their workflows.

The measurements illustrate the possibility to screen three or more component systems for cooperativity or competition within a single 96-well plate smFRET experiment and, by extension, reveal new, unexpected cooperativities and competitions. My general opinion of this study is that a

number of critical steps in the data analysis need to be clarified before the proposed mechanism could be evaluated. I found some of the important conclusions need to be justified, as I detail below.

1. It is not very clear why the multi-well plate is equipped with a heating pad. The authors should provide the rationale behind this and explain it briefly. Also, what exactly would lead to condensation on the well plate sealing that would otherwise be prevented by the heating pad?

2. The authors need to justify the formation of the 'bride population' et al, during the opening and closing dynamics of the Hairpin as a function of salt. Does the 'bride population' refers to a conformation where the Watson-Crick base pairing is on the verge of forming, but not completely formed? How does the distance alter, if the complementary nucleotides are in their respective position?

3. How stable is the conformation associated with the 'bride population'? Is this intermediate close to the open state or to the closed state?

4. Be it is binding of SSB/RecA to DNA or the interaction of Drug molecules with DNA, did the multi-FRET states observed are seen in a single Donor-acceptor pair (in a single trace file) or seen separately in multiple events? The author may present some representative intensity-time traces of the measurement.

5. It is well known that SSB occludes 35 or 65-nt-long stretches on ssDNA and dissolves DNA secondary structures. As SSB is replaced with increasing the concentration of RecA, does it happen to replace one SSB-tetramer at a time? The author may justify if an SSBRecA complex is formed before SSB is been replaced.

6. Is the stage specifically designed to hold a 96-well plate? Or can it be modified to hold plates with a lesser number of holes also?

7. On page 7, were all the experiments and subsequent recordings from the 96 wells done consecutively? What happens to the proteins and DNA that are sampled for such a long duration? Is there any oxygen scavenging system used for the measurements?

8. Apart from the biomolecules under study, oxygen scavengers and triplet-state quenchers are added to the medium. In this case, are these added to all the plates at the same time or sequentially?

9. In Figure 1, what are A01, A02, and so on?

10. Why the instrument is coupled with two lasers, if all the experiments are conducted with one laser only?

11. What information would have been, otherwise missed by the use of a continuous laser?

12. The authors have conducted an experiment wherein a mixture of two DNA constructs that differ in their position of dye labelling (9bp & 21bp) has been loaded into wells and data has been recorded. What was the rationale behind using a mixture of two different DNAs instead of studying them separately? How does it comply with/ how is it relevant to the multi-well confocal setup?

13. In the supplementary file (S6) last paragraph, what does the line "a pipetting time of usually 2-3 h was necessary to load a 96-well plate" imply? Overall, it is not very clear as to how much time it takes to load samples for example in a 96-well plate and how are other things (oxygen scavengers, etc) added.

14. It will be helpful if the authors briefly describe the protocol followed for sample loading and subsequent data acquisition in a single confocal setup and then describe the multi-well setup for readers to understand and appreciate the difference.

Reviewer #4 (Remarks to the Author)

1. In this manuscript, the smFRET-based screening of drug-protein interactions only works when small molecules could induce huge conformation change of proteins. Drug screening is impossible when only minor conformation change occurs with the addition of drugs.

2. The authors stated that the automated multi-well plate assay opens up new possibilities to acquire high-content smFRET datasets for in-depth single-molecule analysis of biomolecular conformations, interactions, and dynamics. How to design multiple conditions to derive the data is expected to be described. Compared with multiple repetitions in a single hole, the diversity of data in this work and the special methods adopted to process the data are expected to be described in detail in the paper.

3. A 96-step gradient of NaCl from 0.1 M to 1 M was employed to explore the salt-dependent opening and closing rates of hpT5. The authors might assume only one conformation exists in solution with NaCl at specific concentration, but actually, multiple conformations may co-exist in solution. How did authors identify their proportion and analyze the transformation process?

4. 32 h is needed to collect extensive data for all provided examples. The authors stated that the addition of a small percentage of surfactant (e.g., Tween20) or a brief preincubation with BSA were employed to avoid loss of fluorescent molecules by non-specific adsorption to the 96-well plate. How did the author consider the photobleaching effect of fluorescent signal in 32 h?

5. In the sentence "Extension of such measurements to larger scale screenings and would benefit massively from an automated multiwell plate format in order to probe multiple small molecules or multiple patient-derived mutations", "and" needs to be deleted.

6. In Figure 2E, Gaussian fitting was employed to find an intermediate state. Can the authors provide R-squared or relevant fitting parameters to show the reliability of fitting results?

RESPONSE TO REVIEWERS' COMMENTS

We thank the editor and the reviewers for their time and efforts to review our manuscript and appreciate the constructive feedback. Below we detail point-by-point our response to the raised comments.

Reviewer 1

In this submission, Hartmann et al. describe an in-house built confocal smFRET instrument capable of automatically collecting data in 96-well plates. They provide a detailed hardware description and open-source software suite, enabling non-experts to assemble the instrument, collect, visualize, and analyze the data. Additionally, they tested their instrument with four cases: a simple rigid DNA FRET construct, hpT5 structural dynamics, protein binding modes, and drug-protein interactions. It took 32 hours to scan 96 wells, but the system can be adjusted for fewer or more wells. To achieve this performance, the authors developed three major additions to a commercial single-molecule confocal machine: an X-Y stage with < 40-micron accuracy, a heating pad on top of the well to prevent condensation, and a liquid dispenser to replenish objective immersion. Although the methodology is sound and there are enough details to reproduce the instrument, the impact on the field remains unclear due to the numerous commercially available instruments, such as the Nikon A1R HD25, Leica TCS SP8, and Zeiss LSM 880 with Airyscan. Overall, I think this is a rigorous instrumental development, but has limited application beyond their own laboratories.

We thank the reviewer for the overall positive assessment and comment below on the specifically raised points.

My detailed comments:

1. What unique features does this in-house built instrument offer that commercial platforms cannot? In fact, Nikon's Ti2-E stage can perform both fine and coarse stepping from nm to cm, outperforming the reported instrument here. Nikon's NIS-elements software also integrates hardware, automation, image processing, and analysis.

***Reply:** We thank the reviewer for this question. It is true that some commercial providers offer fully automated stage scanning for fluorescence microscopy. Yet, to our knowledge, none of the three named microscope manufactures (Nikon, Leica, Zeiss), nor single-molecule spectroscopy manufactures, e.g. like PicoQuant, Oxford Nanoimaging, Exciting Instruments and Abberior Instruments, offer the possibility to perform automated single-molecule FRET experiments.*

In this study, we implemented the use of fully automated multiwell plate scanning for single-molecule FRET measurements in a time-correlated single-photon counting setup. Commercially available instruments are typically built for single-well measurements, but this limitation is overcome with our implementation described here. We used a third-party provider stage, which is offered at a reasonable cost and can be directly controlled with a python script, which simultaneously handles data acquisition of photon streams. In particular, the modular open-source code presented here allows a rapid implementation in different systems without the limitation of using a single commercial provider. Noteworthy, many manufactures, including Physik Instrumente, MadCityLabs, Zaber as well as some microscope vendors, offer Python libraries to address their instruments, e.g. also via Pycro-Manager. Such libraries can be readily included in our autoFRET package to use existing hardware. Additionally, we provide not only data acquisition software, but also the software for data analysis and visualization. We further provide a careful characterization of the precision and accuracy of the presented system. We present showcases of applications ranging from intra-molecular dynamics, protein-protein competitions to drug screening for misfolding diseases.

We added the following lines to the introduction:

“A recent study described high-throughput fluorescence correlation spectroscopy for 96-well plates³⁵, however, to our knowledge the powerful platform of multiwell plates has not yet been transferred to a format suitable for applications in smFRET experiments, neither by manufacturers of microscopes nor by manufacturers of specialized single-molecule spectroscopy instruments.”

2. This platform may be valuable for upgrading existing confocal microscopes without motorized stages. However, fitting an X-Y stage into an existing microscope is case-by-case, and usually, X-Y stages come with their own control software. In this regard, this submission lacks details on how to implement their platform onto a specific model of confocal microscope and appears redundant.

Reply: *We appreciate the comment by the reviewer, however, we believe that the approach we have taken is more general. In fact, the third-party provider used in our example, Zaber, offers various adapters for all major microscope manufacturers, thus it might be easy to find a fitting solution for any given microscope body. Yet, we would like to emphasize that we do not only offer a simple addition of an X-Y stage to a confocal microscope. We also describe the implementation for long term single-molecule FRET measurements, which required a heating pad on top of the 96-well plate, and a water dispenser attached to the microscope (also with adapters for all major commercial microscope provider) and a full software package, i.e. a modular toolbox for instrument control (pyMULTI), data analysis (pyBAT) and data visualization (pyVIZ).*

In order to emphasize the possibility to implement other scanning stages, we added the following paragraph to the SI:

“Of note, we designed our control software pyMULTI in a modular fashion using object-oriented implementation. This allows an easy adaptation to different types of scanning devices through a two-step process: i) Adjustment of the COM port initialization function “initDevices()”; ii) Replacement of the stage movement itself by any device-specific function accepting millimeter scale inputs for x and y direction.”

3. The impact of the 32-hour scanning time lapse from the first sample to the last sample on data quality and analysis is not discussed.

Reply: *We apologize if this was not emphasized in the main text. In Figure S4 and the corresponding supplementary text on page S9, we carefully discuss the effects of long-term measurements including the loss of molecules, but also the instrumental stability like, for instance, the precision and accuracy of FRET values and the correction factors. We have now emphasized this in the results section.*

“In a first set of experiments, we aimed at evaluating the accuracy and precision of multiwell plate measurements and assessing their variability over time. [...] without any significant loss of molecules over the 32-h-measurement period.”

4. The limitation of a 1 ms observation window for many slower biological processes is not addressed.

Reply: *We thank the reviewer for this comment. In fact, there are several techniques available to overcome the temporal limitation in confocal single-molecule FRET experiments, e.g., usage of a bigger confocal pinhole, Recurrence Analysis of Single Particles (RASP), vesicle encapsulation or measurement repeats. In order to discuss the limitations and opportunities, we expanded the conclusion section of our manuscript:*

“Direct extraction of kinetic information from diffusion-based confocal smFRET experiments is typically limited to the millisecond timescale due to the short observation time (i.e., ~1 ms). However, several techniques have been developed to overcome this limitation. These include, for example, the usage of bigger pinholes to expand the confocal volume and thus enable longer observation times, analysis routines that can extract kinetics on longer timescales such as Recurrence Analysis of Single Particles (RASP)⁶, or vesicle encapsulation to slow down diffusion. These techniques can be also applied to our automated approach, thereby extending the extraction of kinetics beyond the millisecond timescale. In fact, in the case of slow equilibration dynamics on the timescale of minutes to hours, the multiwell approach offers the possibility to use the temporal information from well-to-well changes or enables the return to earlier wells for a repeat of conformational sampling. An intermediate time range of seconds to minutes is typically sampled using immobilized molecules combined with microfluidics, which is an attractive, complementary approach^{1,71}. Moreover, our multiwell plate format should also be applicable to fluorescence correlation spectroscopy measurements that enable the extraction of dynamics on the second to nanosecond timescales^{3,72,73}.”

5. For the kinetic information extraction in Figure 2F, it is unclear how the units of k_{on} and k_{off} can be precisely determined to be ms^{-1} . Since single-molecule bursts can only provide the relative distribution of FRET species, as shown in 2E, any relative k_{open} and k_{off} will fit their data for units of any time scale. The reason for precisely pinning the unit to ms^{-1} is not clear.

Reply: *We apologize for the brief discussion in the main text. We describe this in great detail in the SI on page S10 and S11 and the corresponding Figure S5. We expanded the results section and point to details in the SI.*

“The hairpin was designed such that it transiently anneals at a temperature of 25 °C at high salt concentrations, thereby rapidly interconverting between the open and closed conformations on the millisecond timescale, as previously reported³⁹.”

and

“In addition to the open and closed state populations at low and high FRET efficiencies, the hairpin also exhibits a population at intermediate FRET efficiencies, as shown for $[NaCl] = 782$ mM (Fig. 2E and Fig. S5). FRET fluctuation analysis of individual fluorescence bursts (SI and Fig. S5) revealed that this population, termed bridge population, originates from hairpin molecules exhibiting dynamic interconversion dynamics between the open or closed conformation during the ~1-ms-long passage time through the confocal observation volume (i.e., millisecond dynamics)”

6. A discussion comparing this in-house built instrument to the commercial instruments mentioned above would improve the next version of this submission.

Reply: *While commercial scanning confocal microscopes exist, we are not aware of a commercial automated multiwell smFRET approach. Thus, we reason to not include a discussion of differences between such systems. We have included the lack of commercial comparable systems in the introduction, see reviewer 1, point 1.*

7. A discussion of how to integrate this new platform into some specific models to guide non-experts in working on their own models would be helpful.

Reply: *We thank the reviewer for the suggestion. In the supporting information, we illustrated the microscope and the software development in Fig. S1 and S2, listed all the required parts in table S1, and described the setup in detail on page S2 to S4. We furthermore provide the software on github (<https://github.com/SchlierfLAB/autoFRET>) including example data and all*

data presented in this manuscript. We also provide a step-by-step guide on data analysis on the GitHub wiki.

Reviewer 2

The manuscript entitled "Farewell to single-well: An automated single-molecule FRET platform for high-content, multiwell plate screening of biomolecular conformations and dynamics" and authored by Hartmann et al. describes the development of an automated platform for single molecule FRET measurement in a multiwell plate format. This is a notable technical advancement in smFRET study of conformational fluctuations of nucleic acids and protein dynamics on the timescale around or longer than ms. The high-content capability of the platform would be well suited for biomolecular screening and drug discovery.

It should be appreciated that, to demonstrate the effectiveness and evaluate the improvement in precision and accuracy of smFRET measurement using the new platform, the authors carried out comprehensive work on various well-studied systems ranging from the conformational fluctuation of DNA hairpin to protein-drug interaction. The manuscript are generally well-written with detailed descriptions of the materials and methods in supporting information.

We thank the reviewer for the positive assessment of our study.

Below are several specific points for the authors to consider in revision.

1. The significance and the list of the applications of high-content fluorescence imaging in ultrasensitive measurement could be expanded in the introduction. Several platforms have been commercialised. For example, a closely related method - fluorescence correlation spectroscopy - can now be measured in a multiwell plate format.

Reply: We thank the reviewer for this comment and apologize for missing an important reference. We have added high-throughput FCS (Fu et al., 2020) to our introduction.

"A recent study described high-throughput fluorescence correlation spectroscopy for 96-well plates³⁵, however, to our knowledge the powerful platform of multiwell plates has not yet been transferred to a format suitable for applications in smFRET experiments, neither by manufacturers of microscopes nor by manufacturers of specialized single-molecule spectroscopy instruments."

2. Very significant improvement in precision and accuracy in smFRET measurement. But the limitation of the method in terms of the timescale of the conformational kinetics it can measure should be acknowledged. Other related methods which cover different time windows could be discussed.

Reply: We thank the reviewer for this comment. Within this manuscript we do not focus on expanding the technique to measure various intra/intermolecular kinetics, however, there are several analysis techniques available to extract kinetics beyond the 1 ms observation window. We have edited the introduction including references describing how smFRET experiments could cover kinetics from nanoseconds to hours. We have further added a paragraph in the conclusion section describing this limitation and potential solutions.

"Direct extraction of kinetic information from diffusion-based confocal smFRET experiments is typically limited to the millisecond timescale due to the short observation time (i.e., ~1 ms). However, several techniques have been developed to overcome this limitation. These include, for example, the usage of bigger pinholes to expand the confocal volume and thus enable longer observation times, analysis routines that can extract kinetics on longer timescales such as Recurrence Analysis of Single Particles (RASP)⁶, or vesicle encapsulation to slow down

diffusion. These techniques can be also applied to our automated approach, thereby extending the extraction of kinetics beyond the millisecond timescale. In fact, in the case of slow equilibration dynamics on the timescale of minutes to hours, the multiwell approach offers the possibility to use the temporal information from well-to-well changes or enables the return to earlier wells for a repeat of conformational sampling. An intermediate time range of seconds to minutes is typically sampled using immobilized molecules combined with microfluidics, which is an attractive, complementary approach^{1,71}. Moreover, our multiwell plate format should also be applicable to fluorescence correlation spectroscopy measurements that enable the extraction of dynamics on the second to nanosecond timescales^{3,72,73}.”

3. Figure legends are generally too long. They could be made more concise by moving some descriptions to the main text or supporting information.

Reply: We thank the reviewer for this suggestion. We have carefully reduced the length of the figure legends and moved the content as suggested to the main text and the SI.

4. Please quote the standard deviations for " $\langle E_{9bp} \rangle = 0.797$ and $\langle E_{21bp} \rangle = 0.146$ " (in page 7).

Reply: We apologize for the missing confidence interval of the average FRET efficiencies and added the standard error of the mean to the values.

"... $\langle E_{9bp} \rangle = 0.797 \pm 0.001$ and $\langle E_{21bp} \rangle = 0.146 \pm 0.001$,..."

5. Was temperature regulated during measurements? It is not clear at what temperature the effects of salt on the kinetics of DNA hairpin was determined (page 8).

Reply: We apologize for the missing temperature of the DNA hairpin measurement. We performed the experiments at a room temperature of (25.1 ± 0.1) °C. During the measurement, the room temperature was regulated by an air conditioning system including a laminar flow unit. We have added the temperature to the SI table S5.

6. Y-axis title is missing in Fig 2C. Regarding Fig 2F, I would suggest that the x-axis scale starts from 0 instead of 0.1 so the Y-intercepts could be clearly visualized. Also give the reasoning why opening rates under low salt conditions were not included.

Reply: We thank the reviewer for these very good comments, we have added the x-axis title in Fig. 2C (FRET efficiency, E) and formatted the scaling of the x-axis in Fig. 2F. We have excluded the opening rates at low salt condition due to the very low confidence. We have now added them in the revised version, but marked them in grey as being excluded from fitting.

7. A general multiwell plate might not be compatible with confocal imaging. The bottom thickness of the well plate used should be described either in the main text or supporting information.

Reply: We apologize if this information was not presented clear enough. We provide all key components, including the exact 96-well plates, in the SI Table S1. The multiwell plate used in our experiments were #1.5 with a 170 μm thick glass bottom. We have added the thickness to table S1.

Reviewer 3

In the manuscript titled "Farewell to single-well: An automated single-molecule FRET platform for high-content, multiwell plate screening of biomolecular conformations and dynamics", the authors introduced a platform for automated smFRET experiments in a multiwell plate format. With different examples, they have illustrated how accurate and precise FRET efficiencies, as

well as conformational dynamics, molecular competitions, and information on small-molecule-protein interactions, can be obtained from multiwell plate measurements. Along with a detailed description of the hardware components, they have also provided an open-source software suite for data acquisition, analysis, and visualization offering an easily adaptable approach for other labs to integrate the multiwell plate smFRET assay into their workflows. The measurements illustrate the possibility to screen three or more component systems for cooperativity or competition within a single 96-well plate smFRET experiment and, by extension, reveal new, unexpected cooperativities and competitions. My general opinion of this study is that a number of critical steps in the data analysis need to be clarified before the proposed mechanism could be evaluated. I found some of the important conclusions need to be justified, as I detail below.

We thank the reviewer for the overall positive evaluation and clarify open points below.

1. It is not very clear why the multi-well plate is equipped with a heating pad. The authors should provide the rationale behind this and explain it briefly. Also, what exactly would lead to condensation on the well plate sealing that would otherwise be prevented by the heating pad?

Reply: We typically operate with small volumes within the 96-well plates of 100 – 200 μ L and, thus, any evaporation would cause a change in concentrations. To avoid loss of water over 32 h measurements, we have sealed the multiwell plates with Parafilm, yet, we have observed condensation of small water droplets at the Parafilm. The condensation could arise due to a change of the dew point in the sealed chamber, and a slightly lower temperature at the Parafilm compared to the multiwell plate bottom induced by the laminar flow unit of the air conditioning system. To overcome this condensation problem, which might also occur in other labs, we have chosen to add a heating pad operated a few Kelvin above room temperature on top of the Parafilm and did not observe any condensation any more. Measurement stability was high and salt concentration sensitive measurements, e.g. DNA hairpin kinetics, were consistent and reproducible over 32 hours. We have added a short clarification to the SI.

“Without the heating pad, we observed condensation on the seal, likely caused by the increased dew point in the sealed chamber and a slightly reduced temperature at seal, induced by the laminar flow unit of the air conditioning system. After adding the heating pad, we did not observe any condensation, and salt sensitive measurements (e.g., DNA hairpin kinetics) were consistent and reproducible.”

2. The authors need to justify the formation of the ‘bride population’ et al, during the opening and closing dynamics of the Hairpin as a function of salt. Does the ‘bride population’ refers to a conformation where the Watson-Crick base pairing is on the verge of forming, but not completely formed? How does the distance alter, if the complementary nucleotides are in their respective position?

3. How stable is the conformation associated with the ‘bride population’? Is this intermediate close to the open state or to the closed state?

Reply to 2 and 3: We apologize if this information was not presented clear enough. Due to space limitations, we have discussed the identification of the bridge population in the SI. We have added now more information in the results section:

“The hairpin was designed such that it transiently anneals at a temperature of 25 °C, thereby rapidly interconverting between the open and closed conformations on the millisecond timescale, as previously reported³⁹.”

and

“In addition to the open and closed state populations at low and high FRET efficiencies, the hairpin also exhibits a population at intermediate FRET efficiencies, as shown for $[NaCl] = 782 \text{ mM}$ (Fig. 2E and Fig. S5). FRET fluctuation analysis of individual fluorescence bursts (SI and Fig. S5) revealed that this population, termed bridge population, originates from hairpin molecules exhibiting dynamic interconversion dynamics between the open or closed conformation during the $\sim 1\text{-ms}$ -long passage time through the confocal observation volume (i.e., millisecond dynamics).”

With regard to the reviewer’s question about monitoring the formation process of Watson-Crick base pairing: The actual formation process, often termed transition path time, takes place in the case of the DNA hairpin on the order of a few μs or even less (see for instance Neupane et al. PNAS 2017 <https://doi.org/10.1073/pnas.1611602114>). Given our photon density of ~ 117 photons / ms, we do not have in these experiments the time resolution to observe the transition path time. However, continuous instead of pulsed excitation and a photon-by-photon analysis would allow to monitor such time scales also in a multiwell measurement approach.

4. Be it is binding of SSB/RecA to DNA or the interaction of Drug molecules with DNA, did the multi-FRET states observed are seen in a single Donor-acceptor pair (in a single trace file) or seen separately in multiple events? The author may present some representative intensity-time traces of the measurement.

Reply: *In contrast to immobilized molecules, which are often used for SSB or RecA studies, we work completely in solution and record snapshots of populated states from freely diffusing molecules. This means that we do not observe in real time, e.g. the formation of RecA filaments, which typically occurs on the hundreds of milliseconds or even tens of seconds time scale. In our experiments, we sample equilibrium populations at the given SSB, RecA or drug concentration. Changing a single component, e.g. SSB concentration, we change the equilibrium conditions which leads to different populations of the states (cf. Fig. 3B or Fig 3C). Assuming ergodicity (we typically verify prior to long term experiments that we have reached equilibrium), we can extract from our experiments thermodynamic parameters like dissociation constants but cannot access association or dissociation rate constants for such slow processes. In earlier studies of our lab, we have presented intensity time traces of immobilized molecules showing the change of conformation, e.g. for DNA hairpins (Grieb et al. NAR 2017) or for SSB and DnaA (Schärffen et al. Methods 2019). Performing immobilized molecule high throughput experiments is also a very attractive method and we look forward to implementations. We have added a conclusion paragraph discussing these time scales in our manuscript:*

“Direct extraction of kinetic information from diffusion-based confocal smFRET experiments is typically limited to the millisecond timescale due to the short observation time (i.e., $\sim 1 \text{ ms}$). However, several techniques have been developed to overcome this limitation. These include, for example, the usage of bigger pinholes to expand the confocal volume and thus enable longer observation times, analysis routines that can extract kinetics on longer timescales such as Recurrence Analysis of Single Particles (RASP)⁶, or vesicle encapsulation to slow down diffusion. These techniques can be also applied to our automated approach, thereby extending the extraction of kinetics beyond the millisecond timescale. In fact, in the case of slow equilibration dynamics on the timescale of minutes to hours, the multiwell approach offers the possibility to use the temporal information from well-to-well changes or enables the return to earlier wells for a repeat of conformational sampling. An intermediate time range of seconds to minutes is typically sampled using immobilized molecules combined with microfluidics, which is an attractive, complementary approach^{1,71}. Moreover, our multiwell plate format should also be applicable to fluorescence correlation spectroscopy measurements that enable the extraction of dynamics on the second to nanosecond timescales^{3,72,73}.”

5. It is well known that SSB occludes 35 or 65-nt-long stretches on ssDNA and dissolves DNA secondary structures. As SSB is replaced with increasing the concentration of RecA, does it happen to replace one SSB-tetramer at a time? The author may justify if an SSB- RecA complex is formed before SSB is been replaced.

Reply: We thank the reviewer for this very important point. Many studies report extensively about the intriguing complexity of SSB DNA interactions, including also studies by our lab. Adding in a second interaction partner did surprisingly increase the complexity a lot. A first inspection of our data, showed a slight FRET efficiency shift at increasing RecA concentration of the SSB bound state, suggesting an SSB-RecA complex. In order to analyse our equilibrium data (Fig. 3E), we needed to create a multistate model that also includes SSB-RecA complexes (Fig. 3F and SI). We apologize if due to space limitation we have not discussed this more extensively. Briefly, we have optimized the multi-state model parameters (dissociation constants, Hill coefficients) by minimizing χ^2 (SI and Fig. S8 – S10) and have found that a SSB-RecA complex is formed prior to displacement. We have highlighted this complex formation in the results section. In fact, the complexity is highly intriguing and further studies will be necessary to learn more about SSB RecA interactions.

“We found the smallest χ_r^2 value ($\chi_r^2 = 0.108$) for the full reaction scheme involving all 6 states, where $f(E > 0.4)$ describes the combined fraction of state TS_m (SSB₆₅) and TR_qS_m (RecA-SSB) (Fig. S10), an SSB-RecA complex.”

6. Is the stage specifically designed to hold a 96-well plate? Or can it be modified to hold plates with a lesser number of holes also?

Reply: The current software code is written for 96-well plates. It is available as open source (<https://github.com/SchlierfLAB/autoFRET>) and can readily be adjusted to other plate formats. We will also continue to develop the software and regularly integrate updates, possibly also for other plate formats. We have added a note to the conclusion section:

“Of note, the multiwell platform is not limited to 96-well plates but can be readily adjusted to other plate specifications in the open-source code.”

7. On page 7, were all the experiments and subsequent recordings from the 96 wells done consecutively? What happens to the proteins and DNA that are sampled for such a long duration? Is there any oxygen scavenging system used for the measurements?

8. Apart from the biomolecules under study, oxygen scavengers and triplet-state quenchers are added to the medium. In this case, are these added to all the plates at the same time or sequentially?

Reply to 7 and 8: In our experiments we did not add any triplet-state quenchers or oxygen scavengers (all added components are listed in Table S5). As suggested by the reviewer, addition of both components would cause potentially acidification (Swoboda et al. ACS Nano 2012) or ageing of triplet-state quenchers leading to less efficient activity. We did not observe any significant loss of fluorescent molecules (cf. Fig S4A) over 32 h of measurement, likely due to the excitation of only a single well at the time. We have emphasized that we did not use oxygen scavengers or triplet state quencher in the SI Methods section:

“We did not use any oxygen scavengers or triplet quenchers in our buffers.”

9. In Figure 1, what are A01, A02, and so on?

Reply: A01, A02 ... should denote the well number of the 96-well plate. Commercial well-plates are typically labeled with A-H and 01-12 coordinates, which we consistently used in the pipetting tables S7-S11. We have edited the figure legend to clarify.

10. Why the instrument is coupled with two lasers, if all the experiments are conducted with one laser only?

11. What information would have been, otherwise missed by the use of a continuous laser?

Reply to 10 and 11: We have performed our experiments in pulsed-interleaved excitation mode. This means a short (~ 200 ps) direct excitation pulse for the donor fluorophore, followed by a waiting time (~ 20ns) to receive donor and acceptor photons, followed by a short direct excitation for the acceptor fluorophore with a second waiting time (~ 20ns) for acceptor photons. This is a widely used excitation scheme in confocal single-molecule FRET (Hellenkamp, Schmid et al. *Nature Methods* 2018; Agam, Gebhardt, Popara et al. *Nature Methods* 2023) to obtain multi-dimensional information. Hence, besides intensity information of donors and acceptors we obtain also fluorescence lifetime and stoichiometry information of donor and acceptor fluorophores. Stoichiometry means, we calculate a ratio of the photons collected after donor excitation and acceptor excitation (equation 1 in SI) and can by thus sort out molecules. The additional multi-parameter fluorescence detection information (see also Kudryavtsev et al. *Chemphyschem* 2012) can be processed for further filtering and important controls of photophysical artifacts (e.g. fluorophore quenching by protein binding) This is all described in detail in the SI. We added the following part to the main text.

“[...] using pulsed-interleaved excitation (Fig. 1), a widely used excitation scheme in single-molecule fluorescence spectroscopy^{36,37} (see Methods). This allowed us to extract additional information for each detected molecule, such as about stoichiometry and lifetime of the donor and acceptor fluorescence.”

12. The authors have conducted an experiment wherein a mixture of two DNA constructs that differ in their position of dye labelling (9bp & 21bp) has been loaded into wells and data has been recorded. What was the rationale behind using a mixture of two different DNAs instead of studying them separately? How does it comply with/ how is it relevant to the multi-well confocal setup?

Reply: The 9bp & 21bp DNA construct experiment was the first experiment performed with our multiwell setup to demonstrate accuracy and precision of the approach validated on a typical low and high FRET population according to published standards (Hellenkamp, Schmid et al. *Nature Methods* 2018). Furthermore, we used the two FRET populations to determine the correction factors for each measurement repeat, and thereby assessed the measurement stability of 32 h long measurements (page S9 and Fig. S4).

We discuss the findings in the results and the SI. We apologize if this information has not become clear and include a statement in the results section.

“We chose the two DNA constructs to validate the accuracy and precision in extracting low- and high-FRET efficiency populations, according to published standards⁹. Furthermore, we used the two FRET populations to determine correction factors for each measurement repeat, and thereby assessed the measurement stability (SI and Fig. S4).”

13. In the supplementary file (S6) last paragraph, what does the line “a pipetting time of usually 2-3 h was necessary to load a 96-well plate” imply? Overall, it is not very clear as to how much time it takes to load samples for example in a 96-well plate and how are other things (oxygen scavengers, etc) added.

14. It will be helpful if the authors briefly describe the protocol followed for sample loading and subsequent data acquisition in a single confocal setup and then describe the multi-well setup for readers to understand and appreciate the difference.

Reply to 13 and 14: We thank the reviewer for these points. The line “a pipetting time of usually 2-3 h was necessary to load a 96-well plate” should have given the requested information, that it took us by manual pipetting 2-3 h to load samples with the exact solution conditions given in Table S5 – S11.

Briefly, for single-chamber measurements, the preparation of the measurement chamber requires washing and sample loading. After 20 – 30 min, sometimes 1-2 h, the sample is manually removed by pipetting and the chamber washed in a multistep protocol taking approximately 30 min. After starting a new measurement, typically the previous data have to be organized and annotated. Due to the pre-filling of each well with the experimental condition, such a manual sample change is no longer required, allowing overnight experiments at nearly unchanged environment conditions. Additionally, the data management is included in the automated software. This allows us to measure more conditions without delays that would necessitate repeated sample preparations, such as in the case of CFTR transmembrane segments reconstituted in vesicles. Here, phospholipid vesicles fuse and/or aggregate over the course of 1 – 2 weeks requiring repeated vesicle preparation and reconstitution of CFTR transmembrane segments. We have added the following sentences to the manuscript conclusion.

“For confocal smFRET experiments, the measurement of a single condition typically takes about 20 min to 2 h, depending on the system under scrutiny, and is usually followed by chamber cleaning, sample preparation, sample loading and data management, which typically take about 30 min per condition. These laborious and time-consuming steps can be drastically reduced with our multiwell approach. Automatization of data acquisition overcomes repetitive workflows, and multiwell plate handling enables swift sample preparation without repeated preparation steps. Of note, all of the samples described in this work were manually pipetted in an optimized pipetting scheme taking approximately 2–3 h to prepare an entire plate. These sample preparation steps can be further improved.”

Reviewer 4

1. In this manuscript, the smFRET-based screening of drug–protein interactions only works when small molecules could induce huge conformation change of proteins. Drug screening is impossible when only minor conformation change occurs with the addition of drugs.

Reply: We do agree that our assay relies on a change within the protein and in the demonstrated case for CFTR transmembrane domains it is a large conformational change. However, we do not rely on such a large change, and have access to multiple other fluorescence parameters such as anisotropy, diffusion time and fluorescence lifetime which were also used in earlier studies to probe binding. We have added this information to our results section:

“While CFTR TM3/4 shows a relatively large conformational change upon drug binding, employing multiple, complementary fluorescence parameters, as reported earlier⁷⁰, including anisotropy, diffusion time and fluorescence lifetime, can help identifying molecular interactions with rather small conformational changes.”

2. The authors stated that the automated multi-well plate assay opens up new possibilities to acquire high-content smFRET datasets for in-depth single-molecule analysis of biomolecular conformations, interactions, and dynamics. How to design multiple conditions to derive the data is expected to be described. Compared with multiple repetitions in a single hole, the

diversity of data in this work and the special methods adopted to process the data are expected to be described in detail in the paper.

Reply: We thank the reviewer for the comment, and we apologize if the information was missed. All experimental conditions are provided in tables S5 – S11, including the exact concentrations for each well on the plates. The preparation and performance of an automated multiwell plate measurement is described on pages S6 and S7. On our github site (<https://github.com/SchlierfLAB/autoFRET>), we provide the software including an automated installer and a step-by-step guide for data analysis and a primer for data visualization.

With regard to single well measurements, we would like to compare our multiwell approach to a multi-laboratory study (Hellenkamp, Schmid et al. Nat. Methods 2018), which evaluated the accuracy and precision between different laboratories on single well measurements. Our study finds for both DNA rulers mean absolute deviations $\sim 1 \text{ \AA}$, a factor of 2-5 better than what the multi-laboratory study found. We have added more information to the Materials and Methods section:

“Multiwell plate experiments. We loaded our 96-well plates in 2 to 3 pipetting cycles using prepared stock solutions of samples and buffers (Table S5). Detailed concentrations are given for each well in Tables S7 to S11. We describe below briefly the range of conditions.”

3. A 96-step gradient of NaCl from 0.1 M to 1 M was employed to explore the salt-dependent opening and closing rates of hpT5. The authors might assume only one conformation exists in solution with NaCl at specific concentration, but actually, multiple conformations may co-exist in solution. How did authors identify their proportion and analyze the transformation process?

Reply: We thank the reviewer for this comment. Indeed, multiple conformations exist for the DNA hairpin hpT5. In our experiments we observe a closed and open population at high and low FRET efficiencies, as well as an intermediate population. Using a kernel density estimator (S11 and Fig. S5), which analysis the FRET fluctuations within the fluorescence bursts, we found an arc connecting the open and close conformation. These events originate from molecules undergoing conformational changes during the diffusion through the confocal volume, i.e. millisecond dynamics (see also Hartmann, Krainer et al. Molecules 2014). We extracted the opening and closing rate using the 3G model as described in detail in the SI. We apologize that the description was kept very brief in the main text and have edited the main text to point to the more detailed analysis in the SI. We would like to emphasize that the high density of data points at the lower salt concentrations allowed us to model the closing rate dependence on the flexibility of the single-stranded DNA by its change of the apparent concentration (pages S10 to S13).

We have added more details to the manuscript:

“The hairpin was designed such that it transiently anneals at a temperature of 25 °C at high salt concentrations, thereby rapidly interconverting between the open and closed conformations on the millisecond timescale, as previously reported⁶⁹. ”

“In addition to the open and closed state populations at low and high FRET efficiencies, the hairpin also exhibits a population at intermediate FRET efficiencies, as shown for $[\text{NaCl}] = 782 \text{ mM}$ (Fig. 2E and Fig. S5). FRET fluctuation analysis of individual fluorescence bursts (SI and Fig. S5) revealed that this population, termed bridge population, originates from hairpin molecules exhibiting dynamic interconversion dynamics between the open or closed conformation during the $\sim 1\text{-ms}$ -long passage time through the confocal observation volume (i.e., millisecond dynamics).”

4. 32 h is needed to collect extensive data for all provided examples. The authors stated that the addition of a small percentage of surfactant (e.g., Tween20) or a brief preincubation with BSA were employed to avoid loss of fluorescent molecules by non-specific adsorption to the 96-well plate. How did the author consider the photobleaching effect of fluorescent signal in 32 h?

Reply: Due to our experiment design of subsequent measurements of each well (i.e., each well is exposed only to 20 min excitation light), we have not observed any significant amount of photobleaching. We emphasize this in the results section:

“Further, we did not observe any significant loss due to photo bleaching.”

5. In the sentence “Extension of such measurements to larger scale screenings and would benefit massively from an automated multiwell plate format in order to probe multiple small molecules or multiple patient-derived mutations”, “and” needs to be deleted.

Reply: Thank you, corrected.

6. In Figure 2E, Gaussian fitting was employed to find an intermediate state. Can the authors provide R-squared or relevant fitting parameters to show the reliability of fitting results?

Reply: We apologize if this information was not presented clear enough. Due to space limitations, we have discussed the bridge population identification in the SI. We have added some explanation to the results section and pointed to the SI. We discuss extensively fitting results and extracted kinetics in an earlier publication, which we would like to kindly refer the reviewer to: Hartmann, Krainer et al. Molecules 2014. The fitting quality is also reflected in the error bars of the extracted rates (Fig. 2f). Noteworthy, the high amount of data points allowed us also to extract the decrease of the standard deviation of the extrapolated transition rates with increasing data points using a bootstrapping approach (Fig. S5C). We observed a rapid decrease of the standard deviation between 3 and ~ 20 data points, showcasing that already 20-30 data points improve the standard deviation of transition parameters strongly compared to typical manual measurements with 5-10 data points.

REVIEWERS' COMMENTS

Reviewer #2 (Remarks to the Author):

The authors have done a good job in revision and addressed all my comments.

Reviewer #3 (Remarks to the Author):

The authors have provided a satisfactory discussion regarding the points I raised. The current version of the manuscript is written in a fairly straight forward manner with a reasonable logic. Results seem to support the conclusion. The quality of figures and tables seems to be extremely relevant and self explanatory. I recommend accepting the manuscript.

Reviewer #4 (Remarks to the Author):

This work extends and optimizes the means for single-molecule level analysis of conformational changes, interactions, and dynamics of biomolecules. It has certain application value and innovation. The writing is clear and sufficient. The author's reply has clearly answered the questions and doubts I raised, and the quality and readability of the article have been significantly improved.